# Evaluation of lidar-assisted wind turbine control under various turbulence characteristics

Feng Guo[1], David Schlipf[1], and Po Wen Cheng[2]

[1]Wind Energy Technology Institute, Flensburg University of Applied Sciences, Kanzleistraße 91-93, 24943 Flensburg, Germany
[2]Stuttgart Wind Energy (SWE), Institute of Aircraft Design, University of Stuttgart, Allmandring 5b, 70569 Stuttgart, Germany

**Correspondence:** Feng Guo (feng.guo@hs-flensburg.de)

**Abstract.** Lidar systems installed on the nacelle of wind turbines can provide a preview of incoming turbulent wind. Lidar-assisted control (LAC) allows the turbine controller to react to changes in the wind before they affect the wind turbine. Currently, the most proven LAC technique is the collective pitch feedforward control, which has been found to be beneficial for load reduction. In literature, the benefits were mainly investigated using standard turbulence parameters suggested by the IEC 61400-1 standard and assuming Taylor's frozen hypothesis (the turbulence measured by the lidar propagates unchanged to the rotor). In reality, the turbulence spectrum and the spatial coherence change by the atmospheric stability conditions. Also, Taylor's frozen hypothesis does not take into account the coherence decay of turbulence in the longitudinal direction. In this work, we consider three atmospheric stability classes: unstable, neutral, and stable, and generate four-dimensional stochastic turbulence fields based on two models: the Mann model and the Kaimal model. The generated four-dimensional stochastic turbulence fields include realistic longitudinal coherence, thus avoiding assuming Taylor's frozen hypothesis. The Reference Open Source Controller (ROSCO) by NREL is used as the baseline feedback-only controller. A reference lidar-assisted controller is developed and used to evaluate the benefit of LAC. Considering the NREL 5.0 MW reference wind turbine and a typical four-beam pulsed lidar system, it is found that the filter design of the LAC is not sensitive to the turbulence characteristics representative of the investigated atmospheric stability classes. The benefits of LAC are analyzed using the aeroelastic tool OpenFAST. According to the simulations, LAC's benefits are mainly the reductions in rotor speed variation (up to 40%), tower fore-aft bending moment (up to 16.7%), and power variation (up to 20%). This work reveals that the benefits of LAC can depend on the turbulence models, the turbulence parameters, and the mean wind speed.

## 1 Introduction

Traditionally, wind turbine control only relies on the feedback (FB) control strategy. For the above-rated wind operations, the generator speed change caused by the turbulence wind is measured, and the blade pitch is adjusted to maintain the rated rotor/generator speed. This means that the turbine reacts to the wind disturbance only after it has been affected. A nacelle lidar scanning in front of the turbine can provide a preview of the incoming turbulence. Based on the preview, a rotor-effective wind speed (REWS) can be derived and used to provide a feedforward pitch signal. The feedforward pitch signal can be simply added

to the conventional feedback controller (Schlipf, 2015), which is often referred to as lidar-assisted collective pitch feedforward control (CPFF). Apart from CPFF, there are other LAC concepts that have been presented in the literature, e.g. the works by Schlipf et al. (2013b); Schlipf (2015); Schlipf et al. (2020). However, CPFF is so far the most promising technology, and it has been deployed in commercial projects (Schlipf et al., 2018b). Thus, we focus on assessing the benefits of CPFF in this work.

To utilize the lidar measurement for LAC, a correlation study is necessary to determine how much the lidar-estimated REWS is correlated with the actual REWS acts on the turbine rotor. Some facts that could have an impact on the measurement correlation are listed below:

(a) Lidar measurement positions. A typical lidar system has fewer measurement points within the rotor-swept area compared to the rotational sampling rotor. Thus, the lidar-estimated REWS is less spatially filtered.

(b) Line-of-sight (LOS) wind speed $v_{\mathrm{los}}$ measurement, which is the cumulative projection of longitudinal ($u$), lateral ($v$), and vertical ($w$) components in the lidar beam direction. The turbine's aerodynamic performance is mainly driven by the $u$ component, and lidar is expected to measure the $u$ component for control purposes. In reality, the lidar measurements can be contaminated by lateral and vertical wind speed components (Held and Mann, 2019), because of the beam opening angles, the nacelle movement, or the turbine yaw misalignment.

(c) Lidar probe volume. The lidar measurement is the weighted average of LOS along the lidar beam (Peña et al., 2013; Peña et al., 2017).

(d) Turbulence spectrum and coherence. The lidar measurement coherence is mathematically derived based on the spectrum and coherence (Schlipf, 2015; Held and Mann, 2019; Guo et al., 2022a), which will be further discussed in Section 3.

(e) Atmospheric stability. The turbulence spectrum and coherence have been shown to vary by atmospheric stability conditions (Peña, 2019; Guo et al., 2022a).

According to the IEC standard, two turbulence models are commonly used for wind turbine design as provided by the IEC 61400-1:2019 standard, one is the Mann (1994) uniform shear model, and another one is the Kaimal spectra (1972) combined with exponential coherence model (hereafter referred to as Mann model and Kaimal model, respectively). The derivation of lidar measurement coherence based on a specific turbulence model has been studied in the literature. For example, Schlipf et al. (2013a) and Schlipf (2015) show the derivation by the Kaimal model. Mirzaei and Mann (2016), Held and Mann (2019) and Guo et al. (2022a) demonstrate the solution for the Mann model. Based on the two turbulence models, several authors investigated the lidar measurement coherence considering different lidar measurement trajectories and turbine sizes, e.g., the works by Simley et al. (2018), Held and Mann (2019), and Dong et al. (2021). Specifically, in work by Dong et al. (2021), the lidar measurement coherence by the two turbulence models are compared, assuming Taylor's frozen hypothesis. In this paper, we also consider two turbulence models and include turbulence evolution in our analysis.

Once the lidar measurement coherence is analyzed, a filter needs to be designed to filter out uncorrelated information in the lidar-estimated REWS. Because the filter introduces a certain time delay (Schlipf, 2015), a timing algorithm is necessary to ensure the turbine feedforward pitch acts at the correct time. Usually, the time that turbulence requires to propagate from

upstream to downstream, the time delay in the pitch actuator, the time delay by averaging sequential lidar measurements of a full scan, and the time delay caused by filtering should all be considered. In this work, we will contribute by providing a reference lidar-assisted controller. It includes 1) a lidar data processing module that provides the lidar-estimated REWS, 2) a feedforward blade pitch rate provider, and 3) a modified Reference Open Source Controller (ROSCO) with the capability to accept feedforward pitch rate signal. ROSCO (Abbas et al., 2022) is an open, modular, and fully adaptable baseline wind turbine controller with industry-standard functionality.

When evaluating the benefits of LAC, Schlipf (2015) uses the Kaimal model with the turbulence spectral parameters provided by the IEC standard through FAST (Jonkman and Buhl, 2005) (the previous version of OpenFAST (NREL, 2022)) aeroelastic simulation. With a circular scanning lidar, LAC is found to bring a noticeable reduction in the lifetime damage equivalent load (DEL) in the tower base fore-aft bending moment, the low-speed shaft torque, and the blade root out-of-plane moment. However, the variations of turbulence parameters have not been considered.

The recent developments in turbulence simulation tools: *evoTurb* by Chen et al. (2022) and *4D Mann Turbulence Generator* by Guo et al. (2022a) have made it possible to integrate turbulence evolution into aeroelastic simulation. With the updated OpenFAST lidar simulator Guo et al. (2022b), the 4D turbulence field can be imported into OpenFAST, and the upstream lidar measurement can be simulated using the upstream turbulence fields.

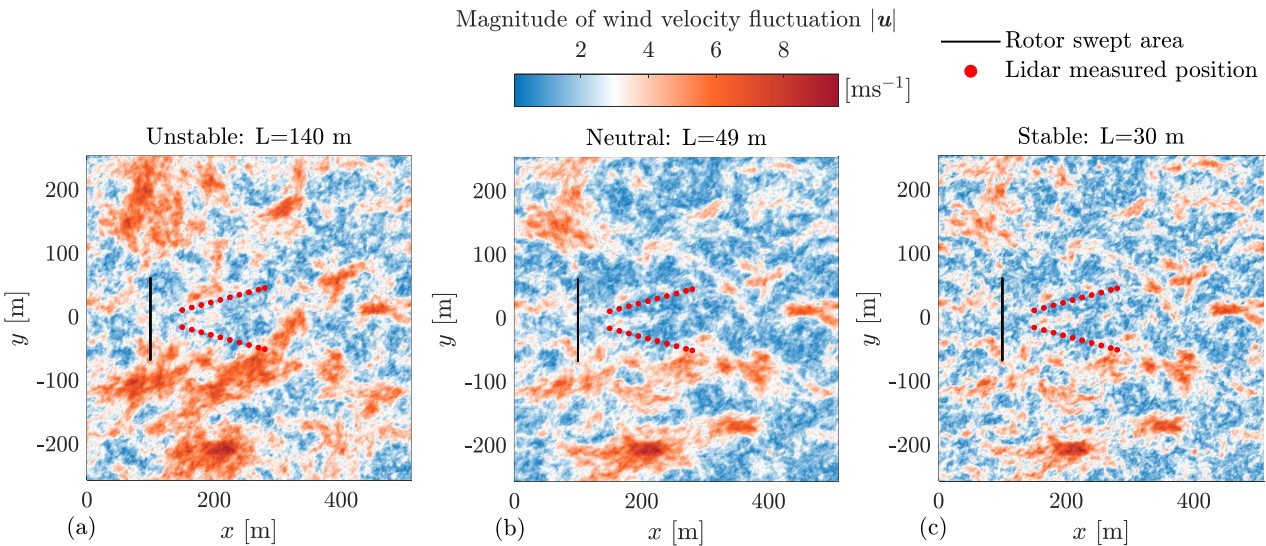

**Figure 1.** Top view of a turbulence field showing the eddy structures under different atmospheric stability, simulated using the *4D Mann Turbulence Generator* with parameters listed in Table 1. The lidar measured positions are plotted based on a typical four-beam pulsed lidar. The rotor swept-area is drawn based on the NREL 5.0 MW reference wind turbine which has a rotor diameter of 126 m. The length scales $L$ are chosen based on studies by Peña (2019) and Guo et al. (2022a).

The variation of turbulence parameters from the standard values given by IEC 61400-1:2019 can be interesting for wind energy. Turbulence parameters under different atmospheric stability classes are investigated and summarized by e.g., Cheynet

et al. (2017), Peña (2019), and Nybø et al. (2020). For example, Figure 1 shows how the turbulence structure changes by the turbulence length scale $L$. A larger coherent eddy structure is observed in the unstable stability, and the eddy structure is much smaller in size under the stable stability. In the neutral case, the eddy structure is somewhere between the two cases. The length scale can have an impact on the power spectrum and turbulence spatial coherence (as later discussed in Section 2.4). Further, the spectrum and coherence can have potential impacts not only on the lidar measurement coherence but also on the turbine loads because the turbulence spectrum peaks can distribute at different frequency ranges, and different frequencies can produce different excitations for the turbine structure motions.

In this work, we summarize how the turbulence spectrum and spatial coherence can vary by atmospheric stability from literature. Three atmospheric stability classes: unstable, neutral, and stable are considered. For each atmospheric stability class, the Mann model parameters are collected, and then the Kaimal model parameters are fitted to have similar spectra and coherence compared to the Mann model. Then the four-dimensional stochastic turbulence fields are generated using *4D Mann Turbulence Generator* (Guo et al., 2022a) and *evoTurb* (Chen et al., 2022). The benefits of LAC are then assessed using a typical four-beam commercial lidar configuration and the 5MW reference wind turbine by NREL (Jonkman et al., 2009) through the lidar simulator-integrated aeroelastic simulation tool: OpenFAST. To compare CPFF with the traditional feedback-only controller, ROSCO is considered to be the baseline feedback controller.

This paper is organized as follows: Section 2 gives the background about turbulence modeling; Section 3 discusses the correlation between the REWS and the lidar-estimated REWS; Section 4 introduces the design of lidar-assisted controller; Section 5 presents and discusses the simulation results; Section 6 draws conclusions for this research.

## 2 Turbulence modeling

In this section, we first introduce the Mann (1994) model and the Kaimal et al. (1972) spectrum and exponential coherence model (Davenport, 1961) used in this work. Then, the methods to include turbulence evolution into the two turbulence models are discussed. Lastly, we show the turbulence spectra and coherence under different atmospheric stability classes.

### 2.1 Mann turbulence model

Mann (1994) model is a spectral tensor model recommended by the IEC 61400-1:2019 standard for wind turbine load calculations. It applies the rapid distortion theory (Hunt and Carruthers, 1990) to an isotropic spectral tensor based on the von Kármán (1948) energy spectrum, to model the shear stretched eddy structures.

At a certain moment, the velocity field can be described by $\tilde{\boldsymbol{u}}(\boldsymbol{x})$, with $\boldsymbol{x} = (x, y, z)$ the position vector in space (Cartesian coordinate). After applying Taylor's frozen hypothesis Taylor (1938) and Reynolds decomposition, the fluctuation part of the turbulence $\boldsymbol{u}(\boldsymbol{x}) = \tilde{\boldsymbol{u}} - \boldsymbol{U}$ about the mean flow $\boldsymbol{U} = (U, 0, 0)$ is assumed homogeneous in space and it can be computed from the Fourier transform

$$\boldsymbol{u}(\boldsymbol{x}, t_0) = \int \hat{\boldsymbol{u}}(\boldsymbol{k}, t_0) \exp(\mathrm{i}\boldsymbol{k} \cdot \boldsymbol{x}) \mathrm{d}\boldsymbol{k}, \tag{1}$$

where $\hat{\boldsymbol{u}}(\boldsymbol{k}, t_0)$ is the Fourier coefficient of the velocity field, i is the imaginary unit and $\int \mathrm{d}\boldsymbol{k} \equiv \int_{-\infty}^{\infty} \int_{-\infty}^{\infty} \int_{-\infty}^{\infty} \mathrm{d}k_1 \mathrm{d}k_2 \mathrm{d}k_3$ means the integration over all the wavenumber vectors $\boldsymbol{k} = (k_1, k_2, k_3)$. Conversely,

$$\hat{\boldsymbol{u}}(\boldsymbol{k}, t_0) = \frac{1}{(2\pi)^3} \int \boldsymbol{u}(\boldsymbol{x}, t_0) \exp(-\mathrm{i}\boldsymbol{k} \cdot \boldsymbol{x}) \mathrm{d}\boldsymbol{x}, \tag{2}$$

with $\int \mathrm{d}\boldsymbol{x} \equiv \int_{-\infty}^{\infty} \int_{-\infty}^{\infty} \int_{-\infty}^{\infty} \mathrm{d}x \mathrm{d}y \mathrm{d}z$. The Fourier coefficients are connected to the elements in the spectral tensor (denoted as $\boldsymbol{\Phi}$) by

$$\Phi_{ij}(\boldsymbol{k})\delta(\boldsymbol{k} - \boldsymbol{k}') = \langle \hat{u_i}^*(\boldsymbol{k}, t_0)\hat{u}_j(\boldsymbol{k}', t_0)\rangle, \tag{3}$$

where $\langle\,\rangle$ means the ensemble average, $^*$ denotes the complex conjugate and $\delta()$ is the Dirac delta function. $\boldsymbol{k}'$ is also the wavenumber vectors and it is used to differentiate with $\boldsymbol{k}$. Equation (3) implies that the ensemble averages of the Fourier coefficients of non-identical wavenumber vectors are all zero. $i, j = 1, 2, 3$ are indexes that stand for $u$, $v$, and $w$ components i.e. $\boldsymbol{u} = (u_1, u_2, u_3) = (u, v, w)$. The detailed expression of $\Phi_{ij}(\boldsymbol{k})$ can be found from the work by Mann (1994). Note that the spectral tensor $\boldsymbol{\Phi}$ is a 3 by 3 matrix for any wavenumber vector $\boldsymbol{k}$ and $\Phi_{ij}(\boldsymbol{k})$ denotes an element in the matrix. Except for the wavenumber vector, there are three other parameters in the model, they are:

— $\alpha\varepsilon^{2/3}$ [m$^{4/3}$s$^{-2}$]: an energy level constant valid in the inertial subrange, composed by the spectral Kolmogorov constant $\alpha$ and the rate of viscous dissipation of specific turbulent kinetic energy $\varepsilon$ (Mann, 1998). This constant actually acts as a proportional gain to the spectral tensor and it is often adjusted to obtain a specific turbulence intensity (TI).

— $L$ [m]: a length scale related to the size of the eddies containing the most energy (Held and Mann, 2019).

— $\Gamma$ [-]: a non-dimensional anisotropy due to shear effect in near-surface boundary layer. When $\Gamma = 0$, the turbulence is isotropic (Mann, 1994, 1998).

Mann (1994) uses $\Gamma$ to calculate the eddy lifetime by

$$\tau(\boldsymbol{k}) = \Gamma \left(\frac{\mathrm{d}U}{\mathrm{d}z}\right)^{-1} (|\boldsymbol{k}|L)^{-\frac{2}{3}} \left[ {}_2F_1\left(\frac{1}{3}, \frac{17}{6}; \frac{4}{3}; -(|\boldsymbol{k}|L)^{-2}\right) \right]^{-\frac{1}{2}}, \tag{4}$$

where ${}_2F_1()$ is a hypergeometric function and $\frac{\mathrm{d}U}{\mathrm{d}z}$ is the mean vertical shear profile. The eddy life time $\tau$ actually distort the wavenumber $k_3$ (corresponds to the $z$ direction) from the initial shearless state $k_{30}$ by $k_3 = k_{30} - \beta k_1$. Here, $\beta = \frac{\mathrm{d}U}{\mathrm{d}z}\tau$ is a non-dimensional distortion factor (Mann, 1994). The effect of the hypergeometric function ${}_2F_1()$ is to have

$$\tau(\boldsymbol{k}) \quad \begin{cases} \propto |\boldsymbol{k}|^{b_1}, & \text{for } |\boldsymbol{k}| \to \infty, \\ \propto |\boldsymbol{k}|^{b_2}, & \text{for } |\boldsymbol{k}| \to 0, \end{cases} \tag{5}$$

where $b_1$ and $b_2$ are two constants standing for the slopes of $\tau$ in logarithmic scale. Instead of using the hypergeometric function. Guo et al. (2022a) proposed another equation for the eddy lifetime

$$\tau(\boldsymbol{k}) = \Gamma \left( \frac{\mathrm{d}U}{\mathrm{d}z} \right)^{-1} \left[ a \left( |\boldsymbol{k}|L \right)^{b_1} \left( (|\boldsymbol{k}|L)^{10} + 1 \right)^{\frac{b_2 - b_1}{10}} \right], \tag{6}$$

$$\text{with} \quad a = \left[ {}_2F_1 \left( \frac{1}{3}, \frac{17}{6}; \frac{4}{3}; -1 \right) \right]^{-\frac{1}{2}}, \tag{7}$$

which is straight forward to adjust the slopes of the eddy-life time. They found that adjusting the slope constant $b_1$ for stable atmospheric stability tends to give better agreements of spectra and coherence between the model and the measurements from a lidar and a meteorological mast. We will use Equation (6) for the rest of this paper.

The one dimensional (along the longitudinal wavenumber) cross-spectra of all velocity components with separations $\Delta y$ and $\Delta z$ can be obtained by

$$F_{ij}(k_1, \Delta y, \Delta z) = \int \boldsymbol{\Phi}_{ij}(\boldsymbol{k}) \exp(\mathrm{i}(k_2 \Delta y + k_3 \Delta z)) \mathrm{d}\boldsymbol{k}_\perp, \tag{8}$$

where $\int \mathrm{d}\boldsymbol{k}_\perp \equiv \int_{-\infty}^{\infty} \int_{-\infty}^{\infty} \mathrm{d}k_2 \mathrm{d}k_3$. Specifically, when $i = j$ and $\Delta y = \Delta z = 0$, it becomes the auto-spectrum of one velocity component at one point, usually written as $F_{ii}(k_1)$. The magnitude-squared coherence between two points in the same $yz$-plane is often interesting which can be calculated by (Mann, 1994)

$$\mathrm{coh}_{ij}^2(k_1, \Delta y, \Delta z) = \frac{|F_{ij}(k_1, \Delta y, \Delta z)|^2}{F_{ii}(k_1) F_{jj}(k_1)}. \tag{9}$$

And the $yz$-plane co-coherence and quad-coherence are defined by

$$\mathrm{cocoh}_{ij}(k_1, \Delta y, \Delta z) = \frac{\Re(F_{ij}(k_1, \Delta y, \Delta z))}{\sqrt{F_{ii}(k_1) F_{jj}(k_1)}}, \tag{10}$$

and

$$\mathrm{quadcoh}_{ij}(k_1, \Delta y, \Delta z) = \frac{\Im(F_{ij}(k_1, \Delta y, \Delta z))}{\sqrt{F_{ii}(k_1) F_{jj}(k_1)}}, \tag{11}$$

where $\Re()$ and $\Im()$ are the real and imaginary number operators, respectively.

## 2.2 Kaimal spectra and exponential coherence model

The Kaimal model given by IEC 61400-1:2019 uses the following formula to determine the auto-spectra of velocity components:

$$S_i(f) = \frac{4\sigma_i^2 \frac{L_i}{U_{\mathrm{ref}}}}{(1 + 6f \frac{L_i}{U_{\mathrm{ref}}})^{5/3}} \tag{12}$$

where $f$ is the frequency, $L_i$ is the integral length scale, $\sigma_i$ is the standard deviation, and $U_{\mathrm{ref}}$ is the reference wind speed equivalent to hub-height mean wind speed. The coherence (with square) of the $u$ components of two points in the $yz$-plane is

described as

$$\gamma_{yz}^2(\Delta yz, f) = \exp\left(-2a_{yz}r\sqrt{\left(\frac{f}{V_{\text{hub}}}\right)^2 + \left(\frac{0.12}{L_{\text{c}}}\right)^2}\right), \tag{13}$$

with $\Delta yz = \sqrt{\Delta y^2 + \Delta z^2}$ the separation distance, $a_{yz}$ the coherence decay constant, and $L_{\text{c}}$ the coherence scale parameter. Note that the coherence without square is used in IEC 61400-1:2019. The $yz$-plane coherence for the $v$ and $w$ components are not given by the IEC 61400-1:2019, and they are ignored in this work.

## 2.3 Modeling of turbulence evolution

The turbulence evolution refers to the phenomenon that the eddy structure changes when the turbulence propagates from upstream to downstream. And it is often represented using longitudinal coherence.

### 2.3.1 Extending Mann model to include evolution

A space-time tensor that extends the three-dimensional Mann spectral tensor $\boldsymbol{\Phi}$ to count for the temporal evolution of the turbulence field has been proposed by Guo et al. (2022a). The space-time tensor is evaluated to provide good agreements on the turbulence spectra and coherence including the spectra of all velocity components and the coherence with longitudinal, vertical-lateral, and all combined spatial separations. The validation has been made using data from a pulsed lidar and a meteorological mast. Details of the model validation can be found in the work by Guo et al. (2022a). The space-time tensor is written as

$$\Theta_{ij}(\boldsymbol{k}, \Delta t) = \exp\left(-\frac{\Delta t}{\tau_e(\boldsymbol{k})}\right) \Phi_{ij}(\boldsymbol{k}), \tag{14}$$

which defines the ensemble average

$$\Theta_{ij}(\boldsymbol{k}, \Delta t)\delta(\boldsymbol{k} - \boldsymbol{k}') = \langle \hat{u}_i^*(\boldsymbol{k}, t_0)\hat{u}_j(\boldsymbol{k}', t_0 + \Delta t)\rangle, \tag{15}$$

where $\hat{u}_j(\boldsymbol{k}', t_0 + \Delta t)$ are the Fourier coefficients of the turbulence field at time $t_0 + \Delta t$. $\tau_e$ is another eddy lifetime (different from $\tau$) that defines the temporal evolution of the turbulence field. The expression

$$\tau_e(\boldsymbol{k}) = \gamma\left[a\left(|\boldsymbol{k}|L\right)^{-1}\left(\left(|\boldsymbol{k}|L\right)^{10} + 1\right)^{-\frac{2}{15}}\right], \tag{16}$$

was found to predicts the longitudinal coherence well as investigated by Guo et al. (2022a). Here, $\gamma$ is a parameter determines the strength of turbulence evolution.

In the space-time tensor, the turbulence field is assumed to travel with a mean reference wind speed $U_{\text{ref}}$. After time $\Delta t$, the field moves downstream in the positive $x$-direction by $U_{\text{ref}}\Delta t$. Thus, for two points with a longitudinal separation of $\Delta x$, the longitudinal coherence (magnitude-squared) of $u$ component can be calculated from

$$\text{coh}_{11}^2(k_1, \Delta x) = \frac{|\int \Theta_{11}(\boldsymbol{k}, \Delta x/U_{\text{ref}})\text{d}\boldsymbol{k}_\perp|^2}{F_{11}(k_1)F_{11}(k_1)}, \tag{17}$$

where

$$F_{11}(k_1) = \int \Phi_{11}(\boldsymbol{k}) \mathrm{d}\boldsymbol{k}_\perp \tag{18}$$

is the auto-spectrum of $u$ component. In practice, the wavenumber-based spectra or coherence is converted to the frequency-based ones using the conversion $k_1 = 2\pi f / U_{\mathrm{ref}}$, assuming Taylor's (1938) frozen hypothesis.

### 2.3.2 Exponential longitudinal coherence model

On the other hand, Simley and Pao (2015) adjusted the exponential coherence model listed in the IEC 61400-1:2019 by replacing the transverse and vertical separations with longitudinal separations, which gives the following expression for the longitudinal coherence

$$\gamma_x^2(\Delta x, f) = \exp\left(-a_x \Delta x \sqrt{\left(\frac{f}{U_{\mathrm{ref}}}\right)^2 + b_x^2}\right), \tag{19}$$

where $a_x$ and $b_x$ are two parameters and $f$ is the frequency. Specifically, $a_x$ determines the decay effect of the coherence and $b_x$ determines the intercept (value at 0 frequency) (Chen et al., 2021). Simley and Pao (2015) validated Equation (19) using Large Eddy Simulations (LES) simulations of different atmospheric stability classes. Besides, Davoust and von Terzi (2016) and Chen et al. (2021) verified the exponential evolution model using lidar measurement, showing the expression by Simley and Pao (2015) agrees well with the measurement. In their study, they found possible $a_x$ and $b_x$ by fitting the coherence calculated from measurement data to the model. As a result, $0 < a_x < 6$ was observed and $b_x$ was found in the order of magnitude $\leq 10^{-3}$.

To include the exponential longitudinal coherence model into the analysis of lidar measurement correlation, a general "direct product" approach is used to combine the lateral-vertical coherence and the longitudinal coherence (Laks et al., 2013; Simley, 2015; Bossanyi et al., 2014; Schlipf et al., 2013a), which means the overall coherence

$$\gamma_{xyz}(f) = \gamma_{yz}(f) \cdot \gamma_x(f). \tag{20}$$

As shown by Chen et al. (2022), the "direct product" approach allows an efficient algorithm to generate the Kaimal model-based 4D stochastic turbulence field using statically independent 3D turbulence fields using *evoTurb*.

### 2.4 Turbulence under different atmospheric stability classes

Atmospheric stability indicates the buoyancy effect on the turbulence generation and it is usually related to the temperature gradient by height. It is interesting to investigate its impact on the filter design of LAC since the turbine will experience different atmospheric stability conditions during operation. The filter is necessary to filter out the uncorrelated frequencies in the REWS estimated by lidar, as will be discussed later in Section 3. In the rest of this paper, we use Mann turbulence parameter sets representative to unstable, neutral, and stable conditions based on the study by Peña (2019) and Guo et al. (2022a), as listed in Table 1. It is worth mentioning that the $\alpha\varepsilon^{2/3}$ parameter is scaled such that the TI corresponds to the IEC 61400-1:2019 class 1A definition. Actually, the turbulence intensity is related to the atmospheric conditions. Usually, TI is generally high

in unstable stability, moderate in neutral stability, and low in stable stability (Peña et al., 2017). In this work, we emphasize analyzing the impact of turbulence length scale and anisotropy on turbine loads and LAC benefits. Therefore, the same TI level is assumed for the three stability classes. This assumption tends to be not realistic, but it helps to identify the impact of length scale on turbine load, as later analyzed in Section 5.2.

As for the Kaimal model, we chose the parameters listed by the IEC 61400-1:2019 for the neutral stability because these
215 parameters were already found to give similar spectra and coherence compared to the Mann model with neutral stability parameters. Also, keeping these parameters allows readers to compare the results with that from existing literature, e.g., Schlipf (2015), Simley et al. (2018), and Dong et al. (2021). For unstable and stable stability classes, we fit the Kaimal spectra by the Mann model-based spectra using the following optimization process:

$$\min_{L_i,\sigma_i} \quad \sum_{n=1}^{N} \left[ \frac{1}{k_{1,n}} \left( S_i(f_n) \cdot f_n - 2F_{ii}(k_{1,n}) \cdot k_{1,n} \right)^2 \right] \;,$$
$$\text{s.t.} \quad k_{1,n} = \frac{2\pi f_n}{U_{\text{ref}}} \quad \text{and} \quad i = 1, 2, 3. \tag{21}$$

Here, $n$ is the index of the discrete frequency vector $f_n$ and wavenumber vector $k_{1,n}$ and $N$ is the size of the discrete vector. Note that the Mann model spectra $F_{ii}(k_{1,n})$ are multiplied by 2 since they are the two-sided spectra while the Kaimal spectra are single-sided. Similarly, we fit the $yz$-plane exponential coherence for the Kaimal model by the Mann model using

$$\min_{a_{yz},L_c} \quad \sum_{n=1}^{N} \left[ \frac{1}{k_{1,n}} \left( \gamma_{yz}(f_n) - \text{cocoh}_{11}(k_{1,n}, \Delta y, \Delta z) \right)^2 \right] \;,$$
$$\text{s.t.} \quad k_{1,n} = \frac{2\pi f_n}{U_{\text{ref}}} \quad \text{and} \quad \Delta y = \Delta z = 20\text{m}, \tag{22}$$

where the fitting uses the co-coherence and ignores the quad-coherence. We fit the co-coherence instead of the magnitude-
225 squared coherence, because the exponential coherence model (Equations 13 and 19) only includes the real co-coherence, whereas the coherence of Mann model includes both co-coherence and quad-coherence. The medium separation $\Delta y = \Delta z = 20$m has been chosen for the optimization problem. For both optimization equations, the squared error in each discrete vector is divided by $k_{1,n}$ to ensure equivalent weighting of the optimization function at a different frequency or wavenumber ranges. The fitted spectra and $yz$-plane coherence are shown by Figure 2(a) and (b), and the turbulence parameters are summarized in
Table 1.

Except for the spectra and $yz$-plane coherence, Guo et al. (2022a) showed that the longitudinal coherence is related to the atmospheric stability based on measurement. In their study, a smaller intercept was found for a more stable class. Also, Simley and Pao (2015) studied the turbulence evolution under different stability classes using LES and the smaller intercept was also observed in stable atmospheric (as shown later in Figure 2). In order to compare the longitudinal coherence under different
atmospheric stability, we use three sets of $\gamma = 200, 400$, and $600$ s to calculate the longitudinal coherence based on the space-time tensor $\boldsymbol{\Theta}$. The reason for choosing these values for $\gamma$ is that they result in coherence close to observations in existing literature, as will be discussed later at the end of this section. Afterward, we fit the exponential coherence (Equation 19) using the following optimization process:

$$\min_{a_x,b_x} \quad \sum_{n=1}^{N} \left[ \frac{1}{f_n} \left( \gamma_x(\Delta x, f_n) - \text{coh}_{11}(k_{1,n}, \Delta x) \right)^2 \right] \;,$$
$$\text{s.t.} \quad \Delta x = 100\text{m}. \tag{23}$$

**Table 1.** The Mann model parameters under different atmospheric stability classes (based on the work of Peña, 2019) and the fitted Kaimal model parameters, calculated using a mean wind speed of 16 ms$^{-1}$. $\alpha\varepsilon^{2/3}$ is scaled such that the TI corresponds to the IEC 61400-1:2019 class 1A definition.

| | Mann | | | Kaimal | | | | | | | |
|---|---|---|---|---|---|---|---|---|---|---|---|
| | $\alpha\varepsilon^{2/3}$ | $L$ | $\Gamma$ | $L_1$ | $L_2$ | $L_3$ | $\sigma_1$ | $\sigma_2$ | $\sigma_3$ | $a_{yz}$ [-] | $L_c$ |
| | [m$^{4/3}$s$^{-2}$] | [m] | [-] | [m] | [m] | [m] | [ms$^{-1}$] | [ms$^{-1}$] | [ms$^{-1}$] | [-] | [m] |
| Unstable | 0.184 | 140 | 2.6 | 744.8 | 181.9 | 126.4 | 2.82 | 2.34 | 1.98 | 6.5 | 1502.0 |
| Neutral | 0.311 | 49 | 3.1 | 340.2 | 113.4 | 27.72 | 2.82 | 2.25 | 1.41 | 12.0 | 340.2 |
| Stable | 0.652 | 30 | 2.4 | 101.1 | 33.3 | 27.0 | 2.82 | 2.26 | 1.83 | 13.1 | 101.1 |

Here we chose to fit the separation at $\Delta x = 100$m, which is the medium separation for a commercial lidar measuring in front of the turbine (Simley et al., 2018; Guo et al., 2022b). The fitted coherence is shown in Figure 2(c). The fitted exponential coherence parameters $a_x$ and $b_x$ are summarized in Table 2, and they show similar trend as the observation by Simley and Pao (2015) using LES. For an unstable atmosphere, $a_x$ is generally larger, and $b_x$ is in a very small order close to 0. In the neutral condition, $a_x$ lies in a medium value, and $b_x$ is also a small order close to 0. As for the stable case, $a_x$ is the smallest, meaning a weaker coherence decay, while $b_x$ is larger, resulting in a smaller intercept.

**Table 2.** The fitted parameters for the exponential longitudinal coherence model.

| Stability | | $\gamma = 200$ s | $\gamma = 400$ s | $\gamma = 600$ s |
|---|---|---|---|---|
| Unstable | $a_x$ | 8.2 | 5.1 | 4.1 |
| | $b_x$ | 8.52 $\times 10^{-5}$ | 8.02 $\times 10^{-5}$ | 7.67 $\times 10^{-5}$ |
| Neutral | $a_x$ | 2.9 | 1.8 | 1.4 |
| | $b_x$ | 1.59 $\times 10^{-4}$ | 1.49 $\times 10^{-4}$ | 1.42 $\times 10^{-4}$ |
| Stable | $a_x$ | 1.6 | 1.0 | 0.8 |
| | $b_x$ | 9.18 $\times 10^{-4}$ | 8.59 $\times 10^{-4}$ | 8.27 $\times 10^{-4}$ |

Based on the study by Guo et al. (2022a), $\gamma$ was found to be 430 s and 207 s for neutral and stable stability classes, respectively, while the value of $\gamma$ in the unstable scenario has not been derived due to a lack of samples from measurement. Chen et al. (2021) performed a probability study of the coherence parameter $a_x$ based on lidar measurement, and it is found to appear between 1 and 2 with a higher probability. According to the analysis by Simley and Pao (2015), $a_x$ tends to be the largest in an unstable condition compared to that in a neutral or stable condition. Based on the previous observations by these authors, and since $\gamma = 200$ s or 400 s gives unrealistically large values of $a_x$ in the unstable atmosphere that are less likely to happen, we decided to choose $\gamma = 600$ s for the unstable condition, which results in $a_x = 4.1$. And $\gamma = 400$ and $\gamma = 200$ are used for neutral and stable stability classes, respectively. In addition, it is worth mentioning that we do not consider the dependence of

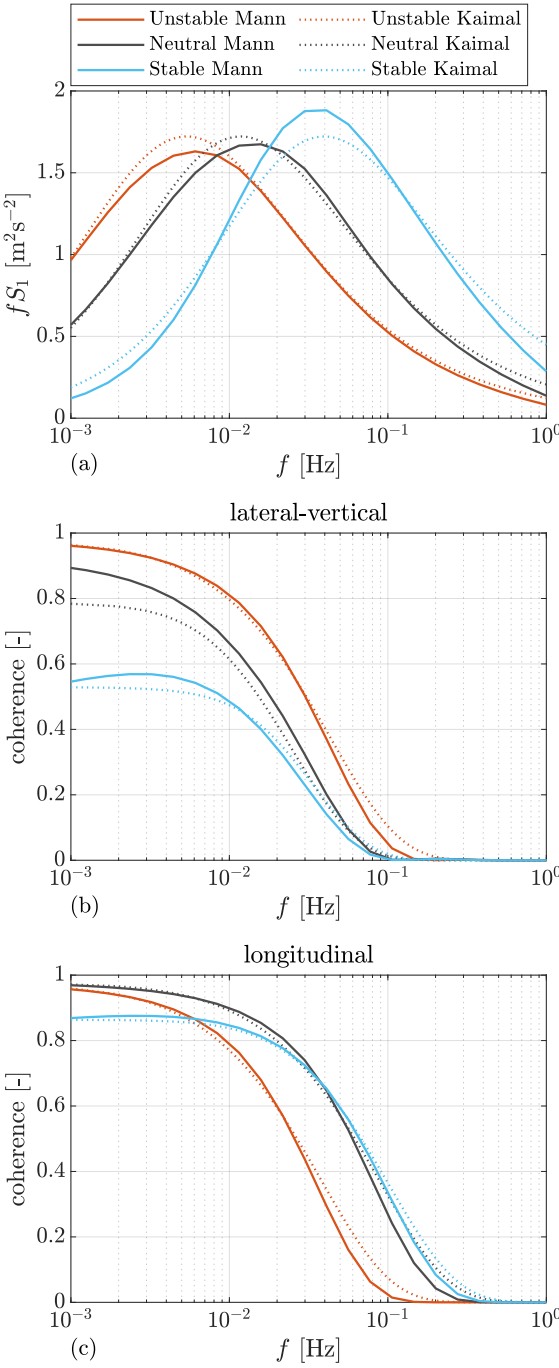

**Figure 2.** (a) The auto-spectra of the longitudinal velocity component under different stability classes. (b) Lateral-vertical coherence of the longitudinal velocity component calculated using the Mann spectral tensor and fitted by the exponential coherence model. Note the co-coherence is shown for the Mann spectral tensor. (c) Longitudinal coherence of the longitudinal velocity component calculated using the space-time tensor and fitted by the exponential coherence model. The results are calculated with a mean wind speed of $16\,\mathrm{ms}^{-1}$.

the turbulence evolution parameters on TI level. The selection of turbulence evolution parameters is based on relevant studies and typical values are chosen. As studied by Simley and Pao (2015), the TI values can be different for the same atmospheric stability, and the evolution parameters show some dependence on the TI values. In the future, a joint probabilistic study on the turbulence spectral parameters, TI levels, and evolution parameters is necessary for defining more realistic simulation scenarios for LAC.

## 3 Correlation between lidars and turbines

In this section, the definitions of REWS and the REWS estimated by lidar will first be discussed. Then the auto-spectra of these two signals and the cross-spectrum between them will be presented. In the end, we summarize the wind preview quality of the investigated four-beam lidar for the NREL 5.0 MW reference turbine under different atmospheric stability classes.

### 3.1 Rotor-effective wind speed

As discussed by Schlipf (2015), one way of defining the rotor-effective wind speed for control purpose is the mean longitudinal component $u$ over the turbine rotor-swept area:

$$u_{\mathrm{RR}}(x) = \frac{1}{\pi R^2} \int\limits_D u(\boldsymbol{x}) \mathrm{d}y \mathrm{d}z. \tag{24}$$

where $D$ denotes the integration over the rotor area defined by rotor radius $R$.

For the Mann model, as derived by Held and Mann (2019), the auto-spectrum of the REWS $u_{\mathrm{RR}}$ can be calculated using the spectral tensor by

$$S_{\mathrm{RR}}(k_1) = \int\limits_{-\infty}^{\infty} \Phi_{11}(\boldsymbol{k}) \frac{4 J_1^2(\kappa R)}{\kappa^2 R^2} \mathrm{d}\boldsymbol{k}_{\perp}, \tag{25}$$

with $\kappa = \sqrt{k_2^2 + k_3^2}$ and $J_1$ the Bessel function of the first kind. The detailed derivation of the auto-spectrum can be found in the works by Held and Mann (2019) and Mirzaei and Mann (2016).

As for Kaimal model, the spectrum is derived by Schlipf et al. (2013a) and Schlipf (2015), i.e.

$$S_{\mathrm{RR}}(f) = \frac{S_1(f)}{n_{\mathrm{R}}^2} \sum_{i=1}^{n_{\mathrm{R}}} \sum_{j=1}^{n_{\mathrm{R}}} \gamma_{yz}(\Delta y z_{ij}, f), \tag{26}$$

where $\Delta y z_{ij}$ is the the separation distance between pint $i$ and $j$ in the same $yz$-plane, and $n_{\mathrm{R}}$ is the total number of points in the rotor area. The detailed derivation of the auto-spectrum can be found in Schlipf (2015).

### 3.2 Lidar-estimated rotor-effective wind speed

Lidar utilizes the Doppler spectrum contributed by the aerosol backscatters within the probe volume to determine wind measurement. It is necessary to include the probe volume averaging effect. Mann et al. (2009) shows that the lidar LOS measure-

ments at a focus position $\boldsymbol{x} = (x, y, z)$ can be approximated by

$$v_{\text{los}}(\boldsymbol{x}) = \int\limits_{-\infty}^{\infty} \varphi(r)\boldsymbol{n} \cdot \boldsymbol{u}(r\boldsymbol{n} + \boldsymbol{x}) \mathrm{d}r, \tag{27}$$

where $\boldsymbol{n} = (n_1, n_2, n_3) = (\cos\beta\cos\phi, \cos\beta\sin\phi, \sin\beta)$ is a unit vector align in the direction of a lidar beam that can be simply calculated after knowing the azimuth angle $\phi$ and elevation angle $\beta$ (see Figure 3 for the definition). $r$ is the displacement along the lidar beam direction from the focused position $\boldsymbol{x}$. $\varphi(r)$ is the weighting function due to the lidar volume averaging. In this work, a typical pulsed lidar is considered whose weighting function is modeled by a Gaussian-shape function (Schlipf, 2015)

$$\varphi(r) = \frac{1}{\sigma_{\text{L}}\sqrt{2\pi}}\exp(-\frac{r^2}{2\sigma_{\text{L}}^2}) \quad \text{with} \quad \sigma_{\text{L}} = \frac{W_{\text{L}}}{2\sqrt{2\ln 2}}, \tag{28}$$

where the full width at half maximum $W_{\text{L}}$ is about 30 m.

Since lidar only provides the wind speed in the LOS direction, the $u$ component is needed to be reconstructed from LOS speed. A simple algorithm is to assume zero $v$ and $w$ components because they usually contribute much less than the $u$ component on the LOS speed. In fact, this is true if lidar beam misalignment to the longitudinal direction is small. Based on this assumption, the lidar-estimated rotor-effective wind speed is often obtained by (see Schlipf (2015))

$$u_{\text{LL}}(t) = \sum_{i=1}^{n_{\text{L}}} \frac{1}{n_{\text{L}}\cos\beta_i\cos\phi_i} v_{\text{los},i}(t), \tag{29}$$

where $n_{\text{L}}$ is total number of lidar measurement positions, $v_{\text{los},i}(x)$ denotes the $i$th lidar measurement position, $\phi_i$ is the azimuth angle of the $i$th measured position, and $\beta_i$ is the elevation angle of the $i$th measured position.

Guo et al. (2022a) suggested to calculate the auto-spectrum of the lidar-estimated REWS ($u_{\text{LL}}$) from the Mann model-based space-time tensor by

$$S_{\text{LL}}(k_1) = \sum_{i,j=1}^{n_{\text{L}}} \sum_{l,m=1}^{3} \frac{1}{n_{\text{L}}^2\cos\beta_i\cos\phi_i\cos\beta_j\cos\phi_j} \int n_{il}n_{jm}\Theta_{lm}(\boldsymbol{k}, \Delta t_{ij})$$
$$\exp(\mathrm{i}\boldsymbol{k} \cdot (\boldsymbol{x}_i - \boldsymbol{x}_j))\hat{\varphi}(\boldsymbol{k} \cdot \boldsymbol{n}_i)\hat{\varphi}(\boldsymbol{k} \cdot \boldsymbol{n}_j)\mathrm{d}\boldsymbol{k}_{\perp}, \tag{30}$$

where $\boldsymbol{x}_i$ and $\boldsymbol{n}_i$ denote the focus position vector and the unit vector of the $i$th lidar measurement respectively, $n_{il}$ is the $l$th element in the unit vector $\boldsymbol{n}_i$, and

$$\hat{\varphi}(\nu) = \int\limits_{-\infty}^{\infty} \varphi(r)\exp(-\mathrm{i}\nu r)\mathrm{d}r = \exp(-\nu^2\frac{\sigma_{\text{L}}^2}{2}) \tag{31}$$

is the Fourier transform (non-unitar convention) of the weighting function of lidar, and $\Delta t_{ij} = (x_i - x_j)/U_{\text{ref}}$ is the time required for turbulence to propagate from position $x_i$ to $x_j$. A more detailed derivation of Equation (30) can be found in the works by Mirzaei and Mann (2016), Held and Mann (2019), and Guo et al. (2022a). In practical lidar data processing for wind turbine control, as discussed in Section 4.2, the lidar measurement data from different measurement gates are phase shifted

to the nearest used measurement range gate using Taylor's (1938) frozen hypothesis. This means that $v_{\mathrm{los},i}(t)$ in Equation 29 should be shifted in time according to the mean wind speed and the longitudinal separation, i.e.

$$u_{\mathrm{LL}}(t) = \sum_{i=1}^{n_{\mathrm{L}}} \frac{1}{n_{\mathrm{L}} \cos\beta_i \cos\phi_i} v_{\mathrm{los},i}\left(t - \frac{x_{\mathrm{nrg}} - x_i}{U_{\mathrm{ref}}}\right), \tag{32}$$

where $x_{\mathrm{nrg}}$ is the longitudinal position of the used measurement range gate nearest to the rotor plane. As a consequence, the phase shifts contributed by longitudinal separations ($x_i$ - $x_j$) in Equation 30 are always zero.

For the Kaimal model, the auto-spectrum can be derived based on the Fourier transform:

$$\begin{aligned} S_{\mathrm{LL}}(f) \quad &= \mathcal{F}\{u_{\mathrm{LL}}\}\mathcal{F}^*\{u_{\mathrm{LL}}\} \\ &= \sum_{i,j=1}^{n_{\mathrm{L}}} \frac{1}{n_{\mathrm{L}}^2 \cos\beta_i \cos\phi_i \cos\beta_j \cos\phi_j} \mathcal{F}\{v_{\mathrm{los},i}\}\mathcal{F}^*\{v_{\mathrm{los},j}\}, \end{aligned} \tag{33}$$

where $\mathcal{F}\{\ \}$ denotes the Fourier transform. The Fourier transform of the $i$th LOS speed $v_{\mathrm{los},i}$ is quite lengthy thus is not extended here. The detailed expression can be found in the work by Chen et al. (2022).

## 3.3 Cross-spectrum between rotor and lidar

When turbulence evolution is considered with Mann model, Guo et al. (2022a) shows that the cross-spectrum between REWS $u_{\mathrm{RR}}$ and the lidar-estimated one $u_{\mathrm{LL}}$ can be calculated using the space-time tensor by

$$\begin{aligned} S_{\mathrm{RL}}(k_1) = &\sum_{i=1}^{n_{\mathrm{L}}} \sum_{j=1}^{3} \frac{1}{n_{\mathrm{L}} \cos\beta_i \cos\phi_i} \int n_{ij}\Theta_{j1}(\boldsymbol{k},\Delta t_i) \\ &\hat{\varphi}(\boldsymbol{k}\cdot\boldsymbol{n}_i)\exp(\mathrm{i}\boldsymbol{k}\cdot\boldsymbol{x}_i - \mathrm{i}k_1 x_i)\frac{2J_1(\kappa R)}{\kappa R}\mathrm{d}\boldsymbol{k}_\perp, \end{aligned} \tag{34}$$

where $\Delta t_i$ is the time required for the turbulence field to move from the $i$th lidar measurement position to the rotor plane, which can be approximated by $\Delta t_i = |\Delta x_i|/U_{\mathrm{ref}}$. Here, $\Delta x_i$ is the longitudinal separation between the rotor plane and the $i$th lidar measurement position and $\Delta x_i = x_i - x_{\mathrm{R}}$, with $x_{\mathrm{R}}$ being he rotor plane position at $x$ axis. For LAC, the lidar measurement data from different range gates are phase shifted to the rotor plane using Taylor's (1938) frozen hypothesis; therefore, this assumption is also made when deriving Equation 34.

Similarly, following Schlipf (2015), the cross-spectrum for Kaimal model is

$$\begin{aligned} S_{\mathrm{RL}}(f) \quad &= \mathcal{F}\{u_{\mathrm{RR}}\}\mathcal{F}^*\{u_{\mathrm{LL}}\} \\ &= \sum_{i=1}^{n_{\mathrm{R}}} \sum_{j=1}^{n_{\mathrm{L}}} \frac{1}{n_{\mathrm{L}} n_{\mathrm{R}} \cos\beta_i \cos\phi_i} \mathcal{F}\{u_i\}\mathcal{F}^*\{v_{\mathrm{los},j}\}, \end{aligned} \tag{35}$$

with $u_i$ the $i$th longitudinal wind component in the rotor swept area. See Chen et al. (2022). for detailed derivation of the Fourier transform of $v_{\mathrm{los},j}$, where the main algorithm is to loop over the Fourier transform of all velocity components included in $u_i$ and $v_{\mathrm{los},j}$.

## 3.4 Lidar wind preview and filter design: case analysis

To evaluate the preview quality of lidar measurement, one can calculate the lidar-rotor coherence by

$$\gamma_{\mathrm{RL}}(f) = \frac{|S_{\mathrm{RL}}(f)|^2}{S_{\mathrm{RR}}(f)S_{\mathrm{LL}}(f)}. \tag{36}$$

Then, a measurement coherence bandwidth (the wavenumber at which the coherence drops to 0.5, noted as $k_{0.5}$) can be found. Note that $k_{0.5} = 2\pi f_{0.5}/U_{\mathrm{ref}}$ where $f_{0.5}$ is the frequency at which the coherence drops to 0.5. $k_{0.5}$ is usually used as the optimization criteria for the LAC-oriented lidar measurement trajectory (Schlipf et al., 2018a).

In this work, we chose the medium-size NREL 5.0 MW reference wind turbine with a rotor diameter of 126 m (Jonkman et al., 2009) and a typical four-beam pulsed lidar trajectory (e.g., WindCube Nacelle and Molas NL). The lidar trajectory is firstly optimized following the method proposed by Schlipf et al. (2018a) using the space-time tensor-based lidar-rotor coherence $\gamma_{\mathrm{RL}}$. The turbulence parameters corresponding to the neutral stability in Table 1 are considered in the optimization process. The optimized trajectory parameters of the used lidar are given in Table 3. A front view of the lidar and turbine geometry is shown in Figure 3.

With the optimized lidar trajectory, we show the coherence $\gamma_{\mathrm{RL}}$ under different stability classes in Figure 4 (a). It can be seen that the coherence using Mann model-based space-time tensor are generally better than that using the Kaimal model. For both models, the coherence in neutral and stable stability classes is higher than that in the unstable stability, which can be caused by stronger turbulence evolution in the unstable situation. The coherence in the unstable case is especially lower using the Kaimal model, which can be caused by the direct product method. Based on the investigation by Simley (2015) using LES, combining coherence using the direct product can underestimate the overall coherence.

Except for the coherence, another indicator of how well the lidar predicts the REWS can be the following transfer function (Schlipf, 2015; Simley and Pao, 2013)

$$|G_{\mathrm{RL}}(f)| = \frac{|S_{\mathrm{RL}}(f)|}{S_{\mathrm{LL}}(f)}. \tag{37}$$

If a filter is designed to have a gain of $G_{\mathrm{RL}}(f)$, it turns out to be an optimal Wiener filter (Simley and Pao, 2013; Wiener et al., 1964), which results produces an estimate of a desired or target signal (here the $u_{\mathrm{RL}}$). The Wiener filter minimizes the mean square error between the target signal and the estimate of the signal. When used for LAC, if the system is modeled as a system with two inputs: REWS and lidar-estimated REWS, and one output: rotor speed, the Wiener filter leads to minimal rotor speed variance as formulated by Simley and Pao (2013). At a certain frequency, the larger gain means that less information needs to be filtered out before the signal is used. So, it indicates how much information measured by the lidar is usable for feedforward control.

The transfer functions under the three investigated stability classes are shown in Figure 4 (b). The transfer function gains are similar in the three stability classes for the space-time tensor-derived results. As for the results by the Kaimal model, the transfer function gain is lower in unstable stability but similar in neutral and stable stability classes.

By the turbulence spectral model, which represents the mean spectral properties, we can obtain the expected Wiener transfer function gain. However, in real operation, the Wiener filter design is more complicated and requires a higher-order filter. In

**Table 3.** Parameters of the optimal four-beam pulsed lidar system. Optimized according to the measurement coherence bandwidth using the space-time tensor model. The definitions of the angles are shown in Figure 3.

| Parameters | Values | Units |
|---|---:|---|
| Number of beams | 4 | [-] |
| Beam azimuth angles $\phi$ | 165.6, 165.6, -165.6, -165.6 | [°] |
| Beam elevation angles $\beta$ | 14.0, -14.0, -14.0, 14.0 | [°] |
| Range gates in $x$ | -50 to -170 | [m] |
| Range gates step in $x$ | 13.3 | [m] |
| Sampling frequency | 1.0 (each beam) | [Hz] |
| Full width at half maximum | 30 | [m] |

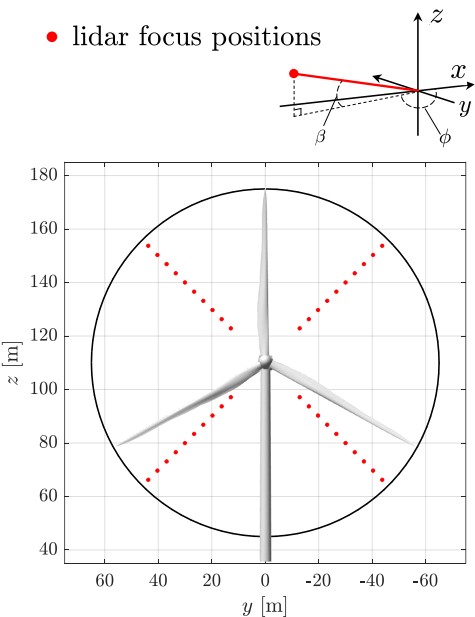

**Figure 3.** The front view of the NREL 5.0 MW turbine and the optimized four-beam trajectory. A reference coordinate system for the lidar system is also shown, where the positive $x$ direction is the mean wind flow direction.

contrast, a linear filter that has similar damping as the Wiener filter can also provide a similar filtering effect as the Wiener filter. The linear filter is usually designed to have a cutoff frequency at -3 dB of the Wiener filter (see Schlipf (2015) and Simley et al. (2018)). The cutoff frequencies as a function of mean wind speed are calculated by fitting the $G_{\mathrm{RL}}$ and are shown in Figure 5. Note that the TI value is also adjusted using the mean wind speed according to the IEC 61400-1:2019 standard. Firstly, both turbulence models indicate that the cutoff frequencies depend on the mean wind speed linearly. Therefore, the cutoff frequency of the filter can be scheduled based on this linearity. Generally, the cutoff frequencies by the Mann model-based space-time

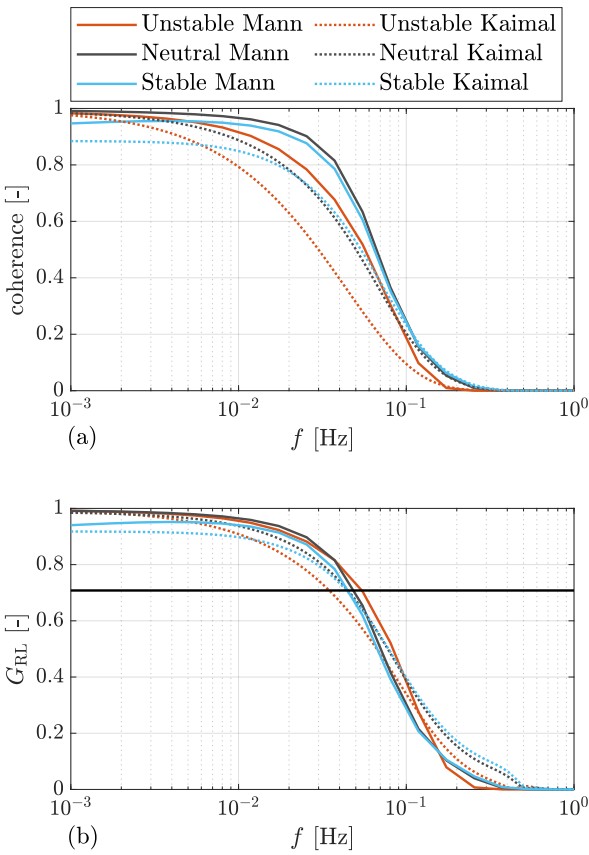

**Figure 4.** (a) Coherence between lidar-estimated RWES and the turbine based REWS. (b) The optimal transfer function gain. The black dot line corresponds to the -3dB magnitude. The results are calculated with a mean wind speed of $16\,\mathrm{ms}^{-1}$.

tensor are generally larger than those by the Kaimal model. For the same turbulence model, the resulting cutoff frequency does not change significantly by the analyzed turbulence stability conditions. The largest difference appear at the highest mean wind speed $24\,\mathrm{ms}^{-1}$ where the difference of cutoff frequency between unstable and stable conditions are about 0.02Hz. As for lower mean wind speed ($\leq 18\,\mathrm{ms}^{-1}$), it can be seen that the turbulence parameters of different atmospheric stability classes do not influence the cutoff frequency very much, and the difference is smaller than $0.01$ Hz. This also indicates that, for mean wind speed $\leq 18\,\mathrm{ms}^{-1}$, the filter design is not very sensitive to the change in turbulence parameters related to atmospheric stability and a constant filter design is robust. In the rest of this work, we will use the constant cufoff frequency derived from neutral stability for both the Mann model-based and the Kaimal model-based simulations. For example, the 0.0490 Hz and 0.0449 Hz will be used respectively for the Mann model and the Kaimal model-based simulations with a mean wind speed of $16\,\mathrm{ms}^{-1}$. However, for a mean wind speed above $20\,\mathrm{ms}^{-1}$, using the cutoff frequency derived from neutral stability is relatively biased

from the cutoff frequency derived for unstable conditions. The impact of this non-ideal filtering should be analyzed further in
future works.

Apart from the case that all measurement gates (see the caption of Figure 5) are considered, another case, where 9 lidar
measurement gates are considered, is also shown in Figure 5. It can be clearly seen that the cutoff frequencies are only slightly
reduced when the first measurement gate is ignored. The reason for considering 9 measurement gates is that the leading time
of the lidar-estimated REWS needs to be larger than the time delays caused by filtering, by time-averaging over full lidar scan,
and by pitch actuator. The leading time of the first measurement gate can be insufficient for very high wind speed, and it must
be ignored. A more detailed discussion about the leading time and time delay will be discussed in Section 4.4.

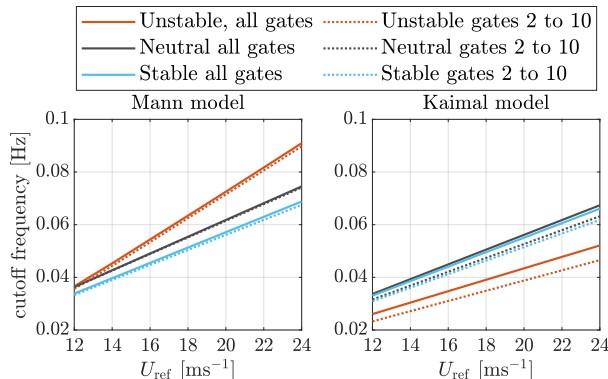

**Figure 5.** The dependency of cutoff frequencies in Hz on the mean wind speed. The cutoff frequency corresponds to -3 dB at the $G_{\mathrm{RL}}$
magnitude. "all gates": the lidar measurement gates from 1 to 10 are considered. "gates 2 to 10": the lidar measurement gates from 2 to 10
are considered.

## 4 Lidar-assisted controller design

In this section, we introduce the lidar-assisted turbine controller theory and its integration into OpenFAST aeroelastic simula-
tion.

### 4.1 Data exchange framework

To configure LAC in the OpenFAST aeroelastic simulation, we chose to use the Bladed style interface (DNV-GL, 2016). The
interface is responsible for exchanging variables between the OpenFAST executable and the external controllers compiled as
Dynamic Link Library (DLL). To make each controller as modular as possible, we programmed an open-source main DLL
(written in FORTRAN), namely the "wrapper DLL". The main function of the wrapper DLL is to call the sub-DLLs by a
specified sequence. Note all the sub-DLLs work based on the same variable exchange pattern specified by the Bladed style
interface. This means each sub-DLL can also be called by OpenFAST independently and directly. Or, several sub-DLLs can be
called by the wrapper DLL together. An overview of the LAC and OpenFAST interface is shown in Figure 6. Three sub-DLLs

will be called by the wrapper DLL following the sequence from up to below in the figure. The source code of a baseline version of these DLLs has been made openly available (see Code Availability).

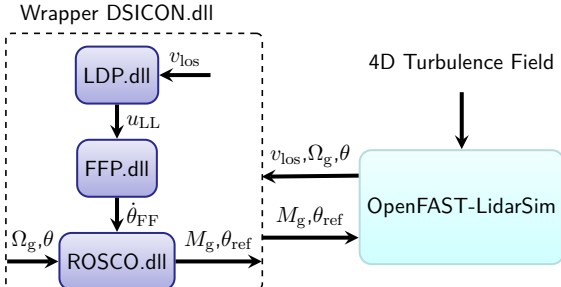

**Figure 6.** The overall OpenFAST and LAC interface. LDP: Lidar Data Processing. FFP: Feedforward Pitch. ROSCO: the reference FB controller.

## 4.2 Lidar data processing

As mentioned before, the lidar measurement data needs to be processed before it can be used for control. The first sub-DLL is the Lidar Data Processing (LDP) which calculates the lidar-estimated REWS from the lidar LOS speed.

In reality, the lidar usually does not measure all beam directions simultaneously. Instead, it sequentially measures from one direction to the next direction. This sequential measurement property is later simulated using the lidar module in the aeroelastic simulation (see Section 5.1.1). Therefore, a time-averaging window needs to be applied to estimate the REWS from a full LOS scan. For the four-beam lidar used in this work, the averaging window is chosen to be 1s which is the time required to finish a full scan by four beams. To apply the averaging window, the LDP module also needs to record the leading time of the successful measurement. The leading time can be approximated by $\Delta x_i / U_{\text{ref}}$. When estimating the REWS, only the LOS measurements whose leading times are within the time-averaging window will be chosen and then Equation (32) is applied to estimate the REWS. Besides, the blade blockage effect is considered in the simulation and this phenomenon is included in the updated OpenFAST lidar module (Guo et al., 2022b). Due to the blade blockage, the LOS measurements for a certain lidar beam are not always available. Therefore, the LDP module estimates the REWS only using all the available LOS measurements.

## 4.3 Feedback-only controller

A typical variable-speed wind turbine is controlled by a blade pitch and generator torque controller. A baseline collective feedback blade pitch control is achieved by a proportional-integral (PI) controller (Jonkman et al., 2009):

$$\theta_{\text{FB}} = k_{\text{p}}(\Omega_{\text{gf}} - \Omega_{\text{g,ref}}) + \frac{k_{\text{p}}}{T_{\text{I}}s}(\Omega_{\text{gf}} - \Omega_{\text{g,ref}}), \tag{38}$$

where $\theta_{\text{FB}}$ is the feedback pitch reference value, $\Omega_{\text{g,ref}}$ is the generator speed control reference, $\Omega_{\text{gf}}$ is the measured and low-pass-filtered generator speed, $k_{\text{p}}$ is the proportional gain, $T_{\text{I}}$ is the integrator time constant, and $s$ is the complex frequency. The

pitch controller is only active above-rated wind speed, and $k_\mathrm{p}$ and $T_\mathrm{I}$ are scheduled to have a constant closed-loop behavior through gain scheduling (Abbas et al., 2022). For the NREL 5.0 MW wind turbine, the desired damping and angular frequency are tuned to be 0.7 and $0.5\ \mathrm{rads}^{-1}$, respectively.

For better code accessibility, the recently developed open-source reference controller: ROSCO (v2.6.0) by Abbas et al. (2022) is used as the reference FB-only controller. ROSCO uses PI controller for the pitch control in the above-rated wind speed operation. In terms of generator torque control in the above-rated operation, we have chosen the option of constant power mode in our simulations, with which the generator torque is set according to the filtered generator speed to keep the electrical power close to its rated value. The generator torque ($M_\mathrm{g}$) is set according to the low-pass-filtered generator speed, the rated electrical power ($P_\mathrm{rated}$), and the generator efficiency ($\eta$) by $M_\mathrm{g} = P_\mathrm{rated}/(\eta \Omega_\mathrm{gf})$. See the work by Abbas et al. (2022) for a more detailed description of the reference controller. We have modified the ROSCO source code to allow it to accept the feedforward pitch rate signal. The feedforward pitch rate (see next section) is added before the integrator of the PI controller.

## 4.4 Combined feedforward and feedback controller

The collective feedforward pitch control proposed by Schlipf (2015) is used in this work where the feedforward pitch reference value is obtained by

$$\theta_\mathrm{FF} = \theta_\mathrm{ss}(u_\mathrm{LLf}), \tag{39}$$

with $u_\mathrm{LLf}$ the filtered REWS estimated by lidar and $\theta_\mathrm{ss}$ the steady-state pitch angle as a function of the steady-state wind speed $u_\mathrm{ss}$. The steady-state pitch curve can usually be obtained by running aeroelastic simulations using uniform and constant wind speed. Figure 7 shows the general control diagram with the lidar-assisted pitch feedforward signal $\theta_\mathrm{FF}$. In practice, the pitch time derivative of the pitch feedforward signal is fed into the integral block of the feedback PI controller. This gives the overall collective pitch control reference as

$$\theta_\mathrm{ref} = \theta_\mathrm{FB} + \frac{1}{s}\dot{\theta}_\mathrm{FF}. \tag{40}$$

A Feedforward Pitch (FFP) sub-DLL is programmed to be responsible for filtering the lidar-estimated REWS and provide feedforward pitch rate at correct time. A first order low-pass filter with the following transfer function

$$G_\mathrm{LPF}(s) = \frac{2\pi f_\mathrm{c}}{s + 2\pi f_\mathrm{c}}, \tag{41}$$

where $f_\mathrm{c}$ is the cutoff frequency as discussed in Section 3.4, is applied to filter the $u_\mathrm{LL}$ signal. Based on the filter cutoff frequency, the time delay introduced by the low-pass filtering of lidar-estimated REWS ($T_\mathrm{filter}$) can be estimated (see Schlipf (2015) for detailed calculation). The pitch feedforward signal is then sent to ROSCO after accounting for the pitch actuator delay ($T_\mathrm{pitch}$), the filter delay, and the half of the time-averaging window ($T_\mathrm{window}$). That is, the signal recorded in the timing buffer that has a time close to the buffer time is activated. The buffer time is defined as

$$T_\mathrm{buffer} = T_\mathrm{lead} - T_\mathrm{filter} - T_\mathrm{pitch} - \frac{1}{2}T_\mathrm{window}. \tag{42}$$

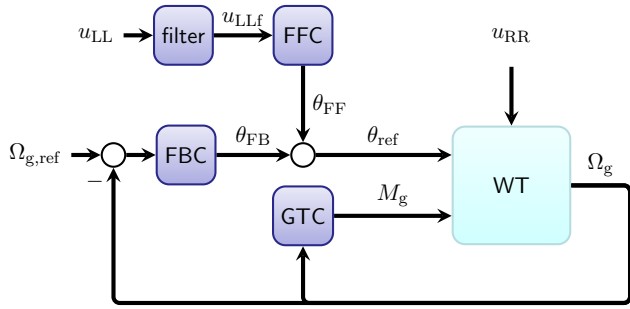

**Figure 7.** The overall control diagram. FFC: feedforward pitch controller, FBC: collective feedback pitch controller, GTC: generator torque controller. Note that the real time pitch angle ($\theta$) signal is also used in the FBC and GTC for controller scheduling.

Here, $T_{\text{window}}$=1 s is the time-averaging window equivalent to one full scan time tof the lidar. It is multiplied by 1/2 in Equation 42, because of the phase delay property of the time-averaging filter (Lee et al., 2018). The actuator delay is chosen to be $T_{\text{pitch}}$=0.22 s based on the phase delay of the pitch actuator. The actuator is modeled as a second-order system with a natural frequency of 1 Hz and a damping ratio of 0.7 (Dunne et al., 2012). Figure 8 shows the leading time ($T_{\text{lead}}$) by the first two measurement gates and the required leading time ($T_{\text{filter}} + T_{\text{pitch}} + \frac{1}{2}T_{\text{window}}$). For the mean wind speed range where the leading time of gate 1 is lower than the required leading time, we only use the lidar measurement gates from 2 to 10 for estimating the REWS. The leading time of gate 2 is sufficient to provide enough leading time for all the considered mean wind speeds.

Another point for the feedforward pitch command is that it is only activated when the REWS is above $14 \text{ ms}^{-1}$. The reason for setting this threshold value is that the pitch curve has much higher gradients with respect to wind speed in the range between $12 \text{ ms}^{-1}$ and $14 \text{ ms}^{-1}$ (Schlipf, 2015), where the turbine thrust is the highest. If the feedforward pitch is activated only depending on the lidar-estimated REWS, a short interval of wind rise or drop in this range can cause a relatively large pitch rate and change in thrust force. Then the benefits of LAC are offset by the additional load caused by these pitch actions.

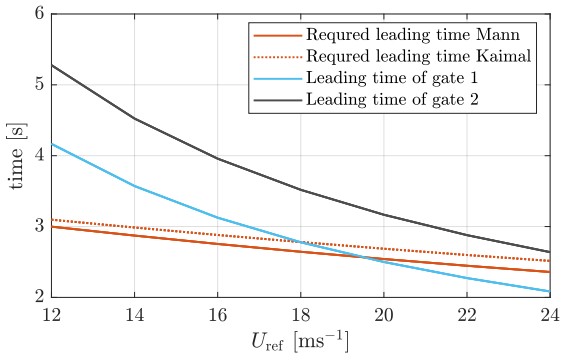

**Figure 8.** The leading time and required leading time for pitch feedforward signal.

# 5    Simulation, results and discussion

In this section, we use the open-source aeroelastic simulation tool OpenFAST to further evaluate the benefits of LAC. The simulation results will be presented and discussed.

## 5.1    Simulation environment

### 5.1.1    Lidar simulation

Previously, the OpenFAST (v3.0) was modified to integrate a lidar simulation module (Guo et al., 2022b). The lidar simulation module includes several main characteristics of nacelle lidar measurement: (a) lidar probe volume, (b) turbulence evolution (lidar measures at the upstream wind field), (c) the LOS wind speed affected by the nacelle motion, (d) lidar beam blockage by turbine blade, and (e) adjustable measurement availability. Based on the study by Guo et al. (2022b) the blade blockage does not have an impact on the lidar measurement coherence for above rated wind speed operation, but special treatment needs to be made to process the invalid measurement caused by the blade blockage effect. In this work, a similar algorithm discussed by Guo et al. (2022b) is used to process the invalid measurement data. Also, the data unavailability caused by low back-scatters is not considered. Therefore, the unavailable data is only caused by the blade blockage.

### 5.1.2    Stochastic turbulence generation

To include the turbulence evolution for the aeroelastic simulation, four-dimensional stochastic turbulence fields are required. We use the newly developed *4D Mann Turbulence Generator* (Guo et al., 2022a) and *evoTurb* (Chen et al., 2022) to generate Mann model and Kaimal model -based 4D turbulence fields, respectively. The turbulence parameters representative for three atmospheric stability classes are used (see Table 1 in Section 2).

For the turbulence field generated by *4D Mann turbulence generator*, since it only contains the fluctuation part of the turbulence, we add the mean field (only for $u$ component) considering a power law shear profile with a shear exponent of 0.2. Each 4D turbulence field has a size of $4096 \times 11 \times 64 \times 64$ grid points, corresponding to the time, and the $x$, $y$ and $z$ directions. The lengths in the $y$ and $z$ directions are both $310\,\mathrm{m}$, which is much larger than the rotor size. The reason for choosing this size is to avoid the periodicity of the turbulence field in $y$ and $z$ directions (Mann, 1998).

For the Kaimal model-based 4D wind fields, *evoTurb* is used, which calls *Turbsim* (Jonkman, 2009) to generated statistically independent 3D turbulence field and then composite 4D turbulence with the exponential longitudinal coherence discussed in Section 2. Only the coherence of $u$ component is considered, and the rest velocity components are not correlated. Similarly, the mean field (only for $u$ component) is considered to be a power law shear profile with a shear exponent of 0.2. Each turbulence field has a size of $4096 \times 11 \times 31 \times 31$ grid points, corresponding to the time, and the $x$, $y$ and $z$ directions. The lengths in the $y$ and $z$ directions are both $150\,\mathrm{m}$, which are enough to simulate the aerodynamic of the $126\,\mathrm{m}$ rotor of the NREL $5.0\,\mathrm{MW}$ turbine. Note that the Kaimal model-based wind fields do not have the issue of periodicity so that the field size is not as large as that of the Mann model-based fields.

For both types of 4D turbulence fields, the time step is chosen to be 0.5 s and the hub height mean wind speed from $12 \text{ ms}^{-1}$ to $24 \text{ ms}^{-1}$ with a step of $2 \text{ ms}^{-1}$ are considered. The turbulence parameters are chosen based on Table 1. However, $\alpha\varepsilon^{2/3}$, $\sigma_1$, $\sigma_2$, and $\sigma_3$ are adjusted according the the mean wind to reach the TI corresponding to class 1A, as specified in IEC 61400-1:2019. The positions in the $x$ direction both contain the rotor plane position and the lidar range gate positions (see Table 3). Taylor's (1938) frozen theory is applied within the probe volume, which has been shown not to influence the lidar measurement spectral properties by Chen et al. (2022). For example, the lidar measurement gate at $x = 50$ m is calculated using the $yz$-plane wind field at $x = 50$ m which is then shifted with Taylor's Frozen theory to count for the lidar probe volume averaging. The time length of each field is 2048 s.

### 5.1.3 Simulation setup

For each stability class, we generate 4D turbulence fields with 12 different rand seed numbers. For each turbulent wind field, the OpenFAST simulation is executed with the following configurations: (a) FB control using ROSCO only; (b) feedforward+feedback (FFFB) control using lidar measurements. All the degree-of-freedoms for a fixed bottom turbine except for the yawing are activated. Each simulation is executed for 31 min. For each simulation, we remove the initial 60 s time series which contains the initialization.

### 5.2 Results and discussion

### 5.2.1 Time series

In Figure 9, we take the one simulation (with a mean wind speed of $16 \text{ ms}^{-1}$) using 4D Mann turbulence with the neutral stability condition as an example to show the time series.

Panel (a) compares the REWS estimated by the lidar data processing algorithm and that estimated by the extended Kalman filter (EKF) (Julier and Uhlmann, 2004) implemented in ROSCO. The lidar-estimated REWS is shifted according to the time buffer by the FFP module so that it does not show any time lag in the plot. The lidar-estimated REWS shows good agreement with that estimated by the Kalman filter. It can be seen that some additional fluctuations with higher frequency appear in the time series of ROSCO-based REWS. This can be caused by the fact that ROSCO only uses one degree-of-freedom model containing the rotor rotational motion and all the other structural motions affecting the rotor speed can be "mistakenly" estimated as wind speed.

Panel (b) shows that the rotor speed obviously fluctuates less using FFFB control compared to that using FB control only. Also, the peak values with FFFB control are smaller.

The tower fore-aft bending moment $M_{\text{yT}}$ is compared in panel (c), where it is generally less fluctuating with the help of LAC. Further, the blade root out-of-plane bending moment ($M_{\text{y,root}}$) is shown by the panel (d), in which FFFB slightly reduces the fluctuation compared to FB-only control. The low-speed shaft torques ($M_{\text{LSS}}$) are compared in panel (e). Again it is clear that the fluctuation with FFFB control is a bit lower than that with FB-only control.

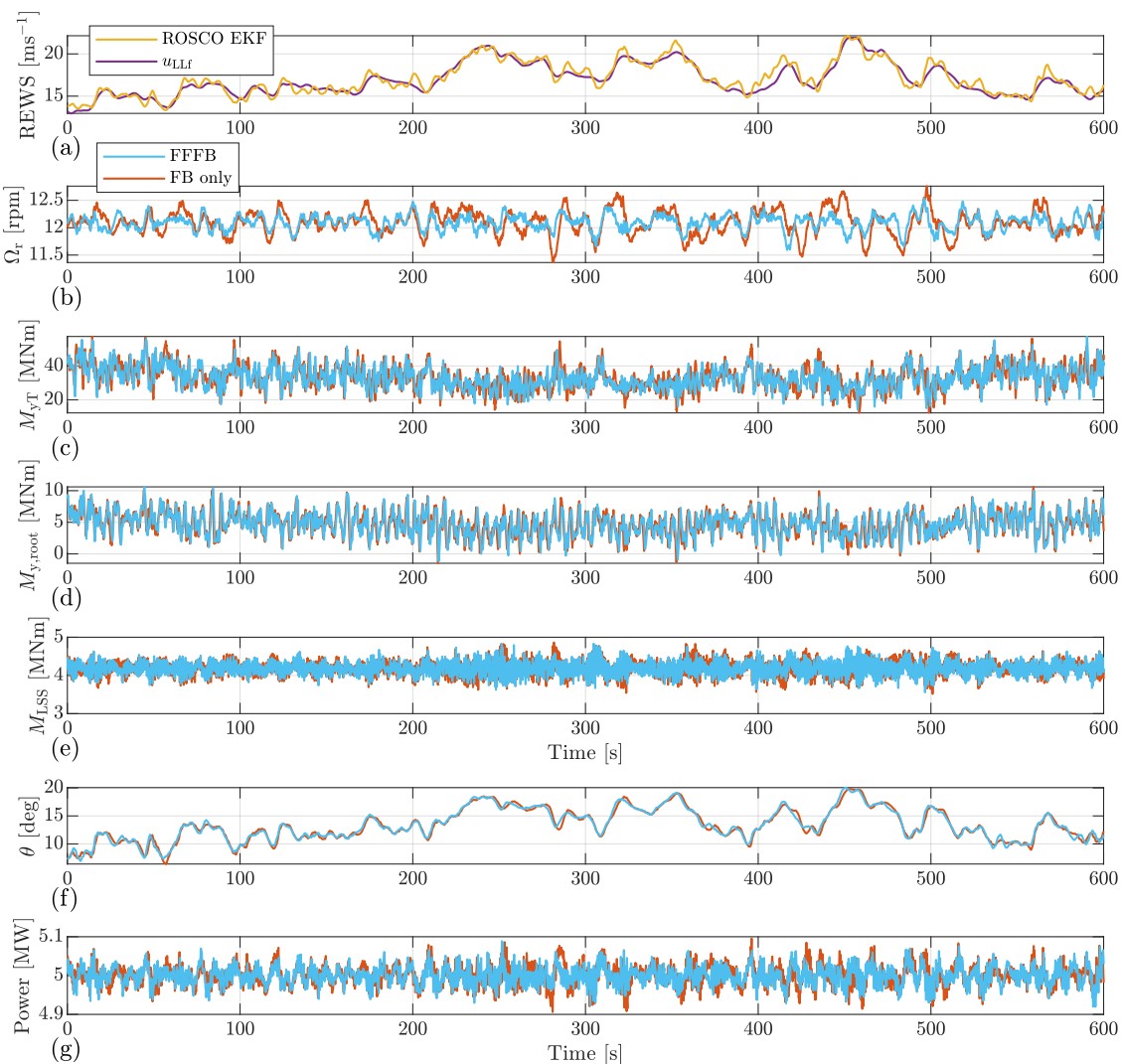

**Figure 9.** The time series collected from OpenFAST simulation. The case with Mann model and neutral stability parameters is shown. Note the same 3D wind field $(y, z, t)$ is applied to the rotor when performing simulations with the FFFB control and the FB-only control. Simulated with a mean wind speed of $16\,\mathrm{ms}^{-1}$. EKF: extended Kalman filter.

In panel (f), we show the pitch action between the two control strategies. The pitch angles in the FFFB control generally lead that by the FB-only control in time, as expected. The pitch angle trajectories are overall similar between the FFFB and FB-only controls.

Lastly, the generator power is shown in panel (g). Here, we can see that the generator power fluctuates even though the constant power torque control mode is activated. The reason is that ROSCO uses low-pass filtered generator speed to calculate the generator torque command by $M_\mathrm{g} = P_\mathrm{rated}/(\eta \Omega_\mathrm{gf})$, as mentioned previously in Section 4.3. If we do not consider the

530 fact that the turbine might have a short interval to go below-rated operation during a wind speed trough, the formula above ensures that the electrical power is constant if the electrical power is calculated using the filtered generator speed. However, the actual electrical power is determined by the non-filtered generator speed, and the difference between the filtered and non-filtered generator speeds determines the power fluctuation. Because the difference is mainly the generator speed fluctuations of high frequencies, we can see that the electrical power contains fluctuations of high frequencies. By comparing FFFB and

535 FB-only controls, it can be seen that reduced low-frequency rotor speed fluctuations are observed in FFFB control. Because the low-frequency power fluctuation is highly coupled with the rotor speed fluctuation (see Panel (b)), less fluctuating power can be expected from the less low-frequency rotor speed fluctuation in FFFB control.

### 5.2.2   Spectral Analysis

We estimate the spectra from the collected time series using Welch's (1967) method. The spectra are averaged by different

samples. Each sample is the aeroelastic simulation result produced by a turbulence field generated by a specific random seed number.

Before comparing the OpenFAST outputs spectra, the spectra of the REWS by the input turbulent wind fields are first compared in Figure 10. Here, the simulated REWS is calculated by averaging the $u$ components within the rotor-swept area from the discrete turbulent wind field. We show that the simulated spectra follow the theoretical ones well, which validates

the turbulence simulation. In Section 2, the single point $u$ component spectrum by the two models is fitted. Also, the $yz$-plane coherence is fitted using a single separation. Here, it can be seen that the REWS spectra by the two models show a similar trend in different atmospheric stability classes. In the unstable case, the REWS spectrum does not reduce a lot compared to a single point $u$ spectrum, and the spectrum peak appears at a lower frequency. This is because the turbulence field has more large-scale coherent structures in the unstable atmosphere, as depicted in Figure 1. In the stable case, everything is opposite to the unstable

case where the REWS spectrum is much lower compared to the single point $u$ spectrum, because of the low-level coherence and the spatial filtering effect of the rotor. In addition, the neutral stability shows a medium spatial filtering effect and the spectrum peak is between that of unstable and stable conditions. For each stability class, it can be seen that the Kaimal-derived REWS generally has a higher spectrum compared to that derived by the Mann model. This can be caused by the fact that the $yz$-plane coherence by the Mann model is more complicated than the exponential coherence model used in the Kaimal model.

Fitting the coherence using one separation is insufficient to represent all possible separations. By comparing the spectra by mean wind speeds of 16 ms$^{-1}$ and 18 ms$^{-1}$, we observe that the spectral peaks are shifted to a higher frequency side in all stability classes.

In Figure 11 and Figure 12, the auto-spectra of some of the most interesting output variables by FB-only control and FFFB control are compared. Figure 11 shows the results using Mann model, and Figure 12 shows the results using Kaimal model.

Panel (a), (b), and (c) compare the rotor speed spectra between FFFB and FB controls under three stability classes. The FFFB control generally reduces the rotor speed spectrum in the frequency range from 0.01 to 0.1 Hz. It can also be seen that the spectra using the Mann model and Kaimal model show some differences, which can be summarized as higher spectra of

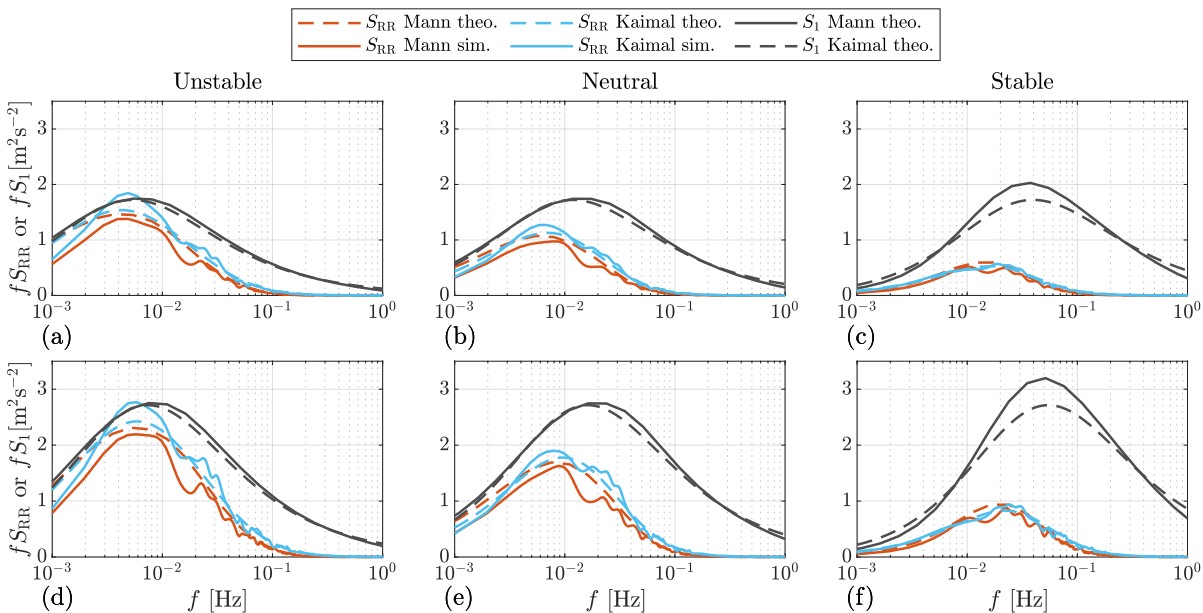

**Figure 10.** The auto-spectra of REWS. "theo.": theoretical spectra by the models discussed in Section 3, i.e. Equation (25) and (26). "sim.": the spectra estimated from the time series of the turbulent wind fields in OpenFAST simulations, using Welch's (1967) method. $S_1$: the auto-spectra of a single point $u$ component. (a) to (c) has a mean wind speed of $16\ \mathrm{ms}^{-1}$. (d) to (f) has a mean wind speed of $18\ \mathrm{ms}^{-1}$.

the rotor motion by the Kaimal model than that by the Mann model. However, the spectra estimated from simulated time series using the two models generally have similar shapes.

The comparison of the tower fore-aft bending moment is shown in panel (d), (e), and (f). In neutral and stable cases, the main benefits bought by FFFB control are the reductions in the frequency range from 0.01 Hz to 0.2 Hz, which is as expected, since the lidar-rotor transfer function (Equation 37) becomes zero close to 0.2 Hz. Below 0.01 Hz, there are not many differences between FB-only and FFFB controls, because the tower fore-aft mode is naturally damped well in this frequency range.

Panel (g), (h), and (i) show the blade root out-of-plane moment of blade 1. There are slight reductions in the blade root
out-of-plane moment in the frequency range from 0.02 Hz to 0.1 Hz contributed by LAC. It can also be seen that the spectrum is mainly composited by the excitation at the 1p (once per rotation) frequency.

The comparison of low-speed shaft torque is shown by the panel (j), (k), and (l). Using FFFB control brings some benefits in the frequency range from 0.01 to 0.1 Hz which is similar to the reduction range of the rotor speed.

Overall, the relative reductions in the spectra bought by adding FF control mainly lie in the frequency range where the
lidar-rotor transfer function is above zero. For very low-frequency ranges, the turbine motions are naturally damped; thus, no obvious benefits are brought by adding the pitch feedforward signal. Based on the spectral analysis, we found reductions significantly in rotor speed, some in tower fore-aft moment, and slightly in low-speed shaft torque. Also, the reductions are observed by both turbulence models in three different atmospheric stability classes.

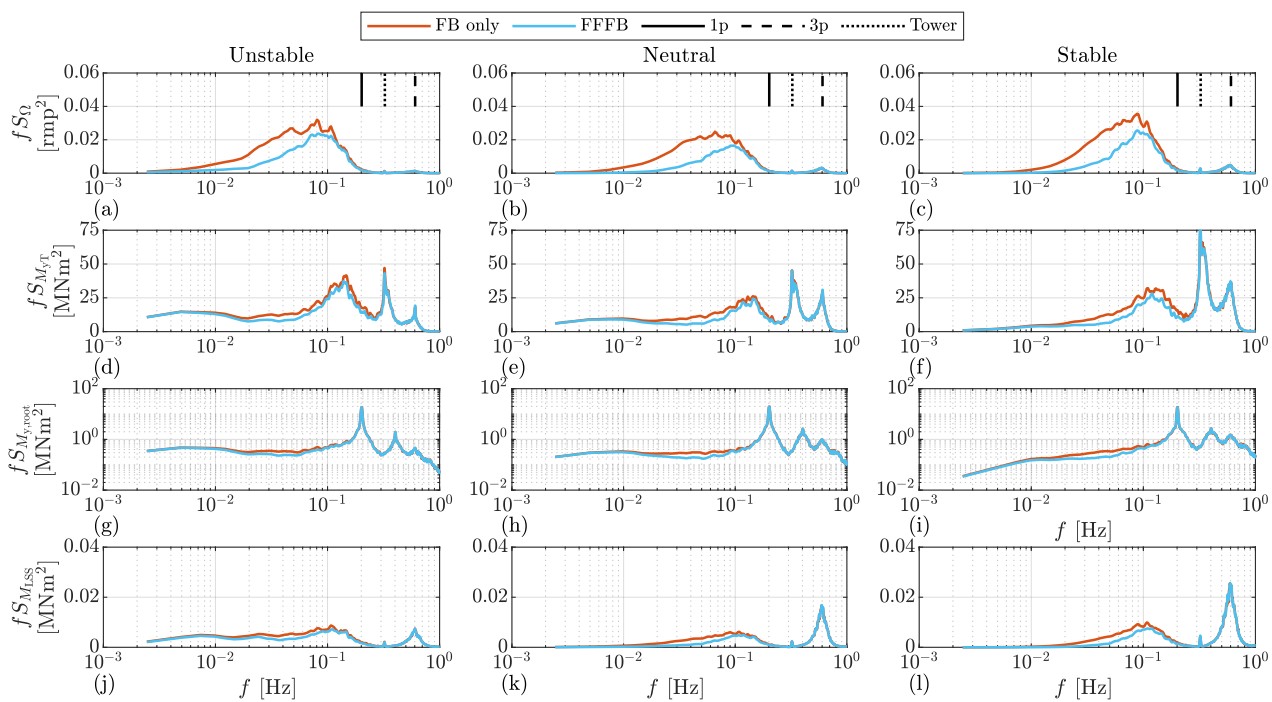

**Figure 11.** The auto-spectra estimated from OpenFAST output time series. The simulation results are obtained using Mann model. The mean wind speed is $16\ \mathrm{ms^{-1}}$. Note that the $y$ axis of the blade root bending moment is set to logarithmic for better readability.

### 5.2.3 Simulation statistic

To further evaluate the benefits of LAC, we calculate the DEL using the rain flow counting method (Matsuishi and Endo, 1968) with $2 \times 10^6$ as a reference number of cycles and a lifetime of 20 years. The Wöhler exponent of 4 is used for the tower fore-aft bending moment and the low-speed shaft torque, and the Wöhler exponent of 10 is used for the blade root out-of-plane bending moment. The averaged DEL is calculated from the results by different random seed numbers. The overall statistics are compared and shown in Figure 13 and Figure 14. For rotor speed, pitch rate, and electrical power ($P_{\mathrm{el}}$) signals, the standard

deviation of time series of each simulation sample is calculated and then the mean value is calculated from all samples. We use the standard deviation of pitch rate (speed) to assess the impact of different control methods on the pitch actuator (also used by Chen and Stol, 2014 and Jones et al., 2018), because pitch speed causes damping torque in the pitch gear and is related to the friction torque of the pitch bearing (Shan, 2017; Stammler et al., 2018) .

     **Mann model-based results**

Figure 13 compares the DEL, standard deviation (STD), and energy production (EP) results by the Mann model. The relative reductions (see the figure caption) between FB-only and FFFB controls are plotted by the grey lines.

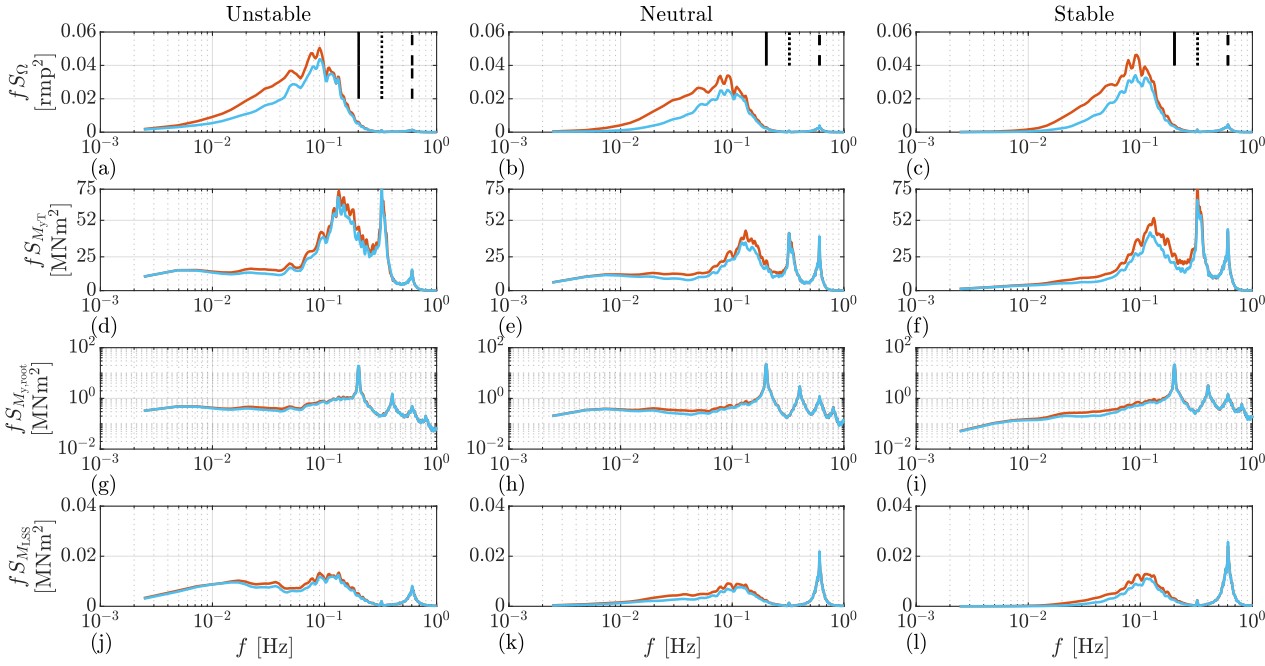

**Figure 12.** The auto-spectra estimated from OpenFAST output time series. The simulation results are obtained using Kaimal model. The mean wind speed is $16 \text{ ms}^{-1}$. Note that the $y$ axis of the blade root bending moment is set to logarithmic for better readability.

There are overall obvious reductions of the tower fore-aft bending moment DEL in all the investigated atmospheric stability classes. The largest reduction is found to be 16.7% by a mean wind speed of $22 \text{ ms}^{-1}$ and under an unstable atmosphere. In the unstable case, it can be seen that the reduction is more clear with a higher wind speed. On the opposite, for the stable stability, the reduction is larger at $16 \text{ ms}^{-1}$ and $18 \text{ ms}^{-1}$, and it reduces as wind speed increases. As for the neutral case, the benefits are the greatest close to $18 \text{ ms}^{-1}$. However, with the mean wind speeds below $14 \text{ ms}^{-1}$ and in the unstable and neutral cases, the FFFB benefits becomes marginal. This can be caused by a higher possibility to pass the wind speed range where the feedforward pitch is inactivated, as discussed in Section 4.4.

As for the low-speed shaft torque, the DEL is reduced by more than 4.0% under the unstable case for wind speed above $18 \text{ ms}^{-1}$. In addition, the reduction is about 1.5-3.3% and 1.4-2.3% under neutral and stable cases, respectively.

The DEL of the blade out-of-plane moment is reduced by introducing LAC. More benefits (about 2.7-6.0%) are found under the unstable case. In the neutral stability, the reduction is better at $20 \text{ ms}^{-1}$, where the value is close to 4.3%, and it drops to 2.5% by higher wind speeds and to 1.3% by lower wind speeds. As for stable atmosphere, the reduction is more obvious (around 3.0%) at wind speeds between $16 \text{ ms}^{-1}$ and $20 \text{ ms}^{-1}$.

The STD of rotor speed is found to be reduced significantly using FFFB control. The reductions are more than 20% and up to 40%. Also, it can be seen the reductions are more significant under higher mean wind speeds, which is similar in all the three-atmosphere stability classes.

Introduction the FF pitch also generally helps to reduce the standard deviation of pitch rate (speed) $\dot{\theta}$. Among the three stability classes, the standard deviations of pitch rate are reduced clearly (vary from 2.0% to 6.1%) from 14 ms$^{-1}$ to 20 ms$^{-1}$. However, the reduction stops at the mean wind of 24 ms$^{-1}$ for unstable and neutral conditions. In the stable atmosphere, the pitch rate STD only reduces with mean wind speeds smaller than 20 ms$^{-1}$.

As for the electrical power STD, it is reduced obviously by about 16% in the unstable case for wind speed above 18 ms$^{-1}$, by about 17% in the neutral case for wind speed above 16 ms$^{-1}$, and 13% in the stable case for wind speed above 14 ms$^{-1}$.

With the same mean wind speed but under different stability cases, the electricity productions are similar either using LAC or not. For all the stability conditions, the electricity productions are lower at wind speeds below 14 ms$^{-1}$ because there is a higher probability that the REWS goes below the rated value and the electrical power does not reach the rated power.

**Kaimal model-based results**

The results using the Kaimal model are shown in Figure 14. Generally, under different stability classes and mean wind speeds, the statistics show a similar trend as the results obtained by the Mann model. However, the values show some differences.

In terms of tower fore-aft bending moment, the reductions of DEL are from 10.4% to 13.4% with a mean wind speed from 18 to 20 ms$^{-1}$ under unstable and neutral conditions. In the stable case, the reduction is close to 11.5% with the mean wind speed of 16 ms$^{-1}$ and it drops wither higher mean wind speeds.

The results of low-speed shaft DEL show a similar trend to that using the Mann model. On average, for wind speed above 16 ms$^{-1}$, the shaft load is reduced by around 2.3%, 1.9%, and 1.7%, respectively, under the three investigated stability classes.

Generally, the reduction of the blade root load simulated using the Kaimal model is similar to that based on the Mann model. On average, for wind speed above 16 ms$^{-1}$, the blade root DEL is reduced by around 4.1%, 3.0%, and 3.0%, respectively, under the three investigated stability classes.

The STD of rotor speed is found to be reduced obviously using FFFB control. The reductions are more than 15% and are up to 30%. The result shows a similar trend to that of the Mann model-based result. However, we can also see the reduction is less than that shown by the Mann model.

The pitch actions show high similarity with that simulated using the Mann model. At mean wind speeds from 16 ms$^{-1}$ to 20 ms$^{-1}$, the reductions in pitch rate STD are about 3.0% to 3.5% under unstable and neutral stability classes and they become less in other mean wind speeds. For the stable case, the reduction is higher at 16 ms$^{-1}$, reaching 6.2%, but decreases rapidly as the mean wind speed increases. For very high mean wind speeds above 22 ms$^{-1}$, the pitch rate STD is increased using LAC.

Since the variation in electrical power is highly linked with the rotor speed. The reductions in the STD of power lie around 10%, 13%, and 11%, respectively, under the three investigated stability classes. These values are smaller than that observed using the Mann model.

The electricity production shows very similar results to that simulated by the Mann model. Using LAC has a marginal impact on electricity production.

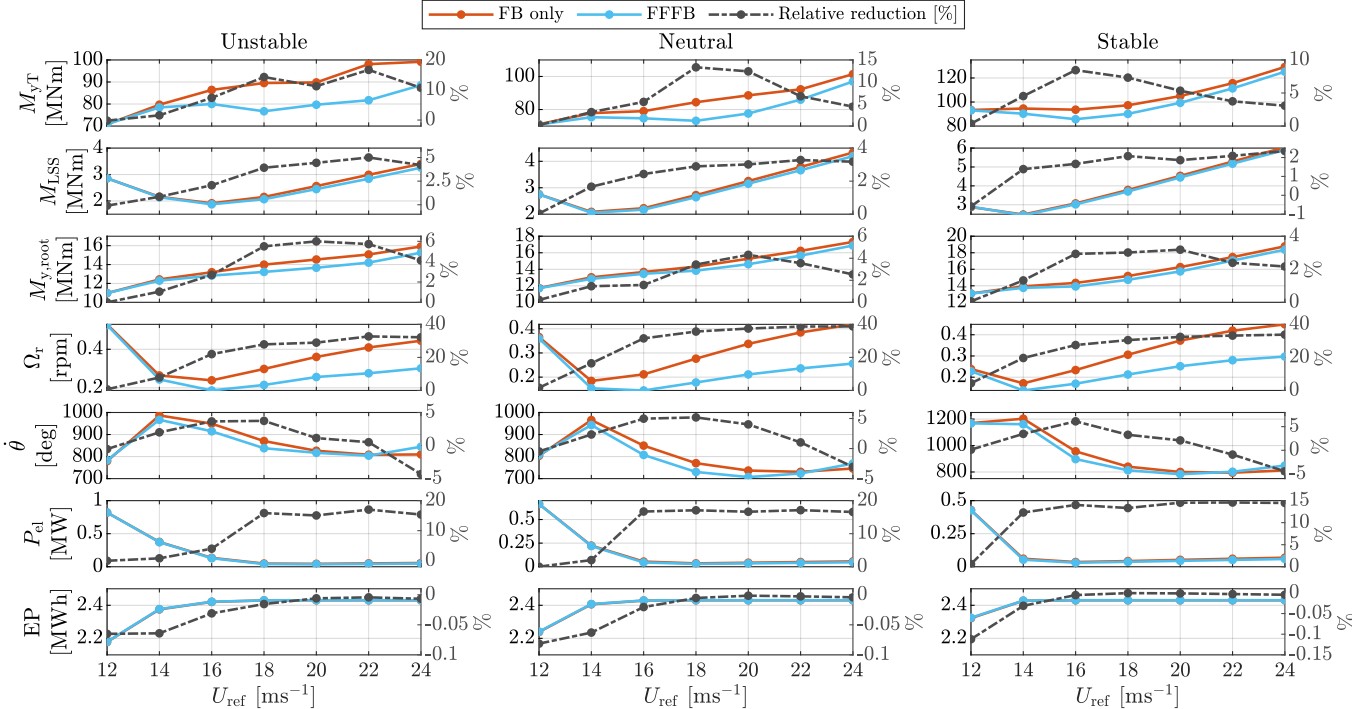

**Figure 13.** Comparison of DEL ($M_{\mathrm{yT}}$, $M_{\mathrm{LSS}}$, $M_{\mathrm{y,root}}$), STD ($\Omega_{\mathrm{r}}$, $\dot{\theta}$, $P_{\mathrm{el}}$), and EP, simulated using Mann model. Note that the value of the relative reduction are reflected by the right side $y$ axis. Relative reduction: (FB-only-FFFB)/(FB-only).

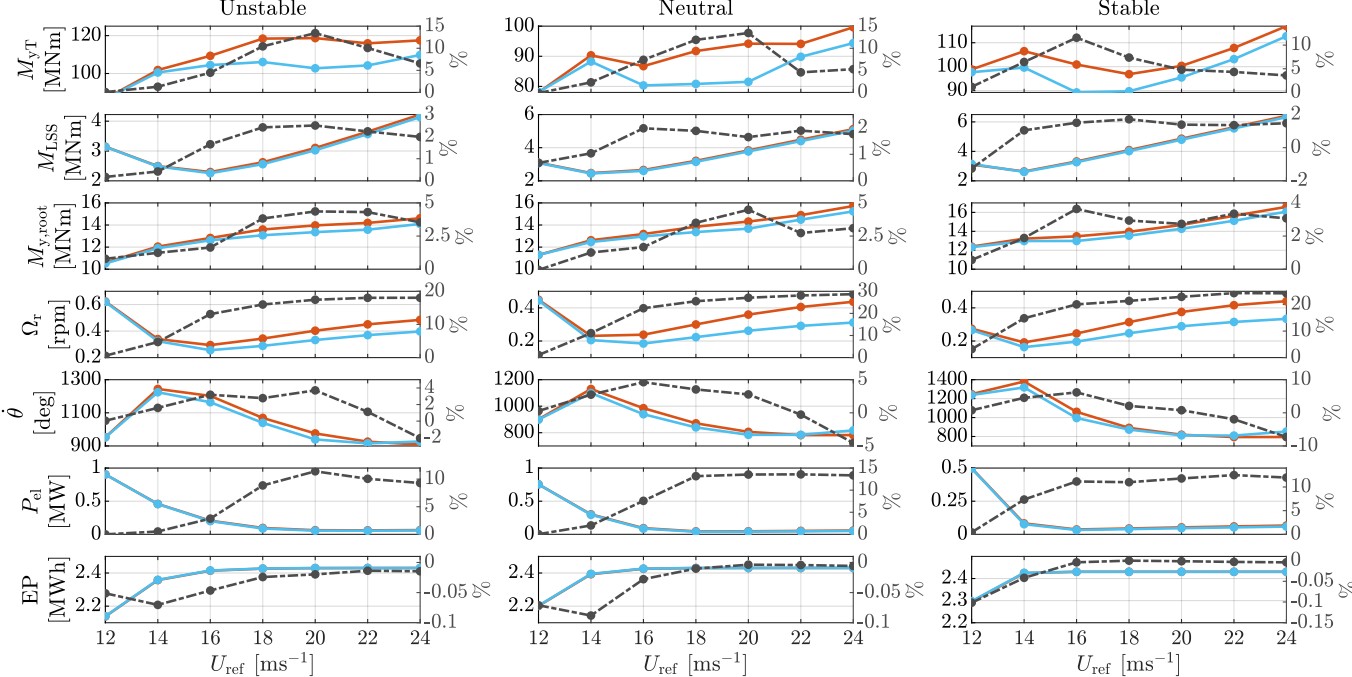

**Figure 14.** Comparison of DEL ($M_{\mathrm{yT}}$, $M_{\mathrm{LSS}}$, $M_{\mathrm{y,root}}$), STD ($\Omega_{\mathrm{r}}$, $\dot{\theta}$, $P_{\mathrm{el}}$), and EP, simulated using Kaimal model. Note that the value of the relative reduction are reflected by the right side $y$ axis. Relative reduction: (FB-only-FFFB)/(FB-only).

In general, the benefits of LAC in load reduction by a four-beam lidar are clear. However, we also show that there are some uncertainties and differences when assessing LAC by different IEC turbulence models. Among the compared turbine loads, LAC has the most significant load reduction effect in the tower base fore-aft bending moment. There are also considerable reductions in speed and power variations. The electrical power generation is not significantly affected by introducing LAC. The load reductions also show differently under different turbulence parameters represented by different atmosphere stability

classes. For different stability conditions but the same mean wind speed, it can be seen that the LAC benefits for the load reduction are overall highest in the unstable, medium in neutral, and lowest in stable atmospheric classes. The reason could be the difference in turbulence length scales. The turbulence length scale is lower under a stable condition which means the peak of the turbulence spectrum appears at a higher wavenumber/frequency (based on the conversion $f = k_1 U_{\mathrm{ref}}/2\pi$). The turbine's structural loads are mainly excited by frequency above 0.1 Hz, e.g. the tower natural frequency, the shaft natural frequency

(above 1 Hz), the 1p frequency and the 3p (three times per rotation) frequency. If the spectrum has a higher peak frequency, the load will be more dominated by the higher frequency parts due to the higher excitation of the natural modes. Then the LAC benefits become less significant because it mainly reduces the loads below 0.1 Hz (for the lidar and turbine we used). When considering different mean wind speeds, the discussions above indicate that a higher mean wind speed shifts the spectral peak frequency to be a higher value; therefore, the LAC benefits become less. For the stable condition, the spectral peak frequency

is naturally high due to the smaller turbulence length scale so it is more sensitive to the changes in the mean wind speed. For unstable and neutral cases, the spectrum peak frequency is naturally lower than that in the stable condition, thus the LAC benefits do not decrease as fast as that in the stable condition.

## 6 Conclusions

This paper evaluates lidar-assisted wind turbine control under various turbulence characteristics using a four-beam liar and the

660 NREL 5.0 MW reference turbine. The main contributions of this work include: (a) summarizing the turbulence spectra and the coherence under various atmosphere stability conditions, (b) analyzing the requirement of filter design for lidar-assisted wind turbine control under various turbulence characteristics, (c) developing a reference lidar-assisted control package, and (d) evaluating the benefits of lidar-assisted wind turbine control using two turbulence models through aeroelastic simulations.

  Currently, two turbulence models (the Mann model and the Kaimal model) are provided by the IEC standard for turbine

aeroelastic simulation. The recent research has made it possible to generate 4D stochastic turbulence fields in aeroelastic simulation for both the Mann model and Kaimal model, which allows for simulating lidar measurements more realistically and assessing the potential benefits by lidar-assisted control more reasonably. When evaluating the benefits of lidar-assisted control, previous research uses the Kaimal model with fixed turbulence spectral parameters provided by the IEC standard Schlipf (2015). Thus, the variations of turbulence characteristics by atmospheric stability have not been considered. In this

study, we defined three turbulence cases whose characteristics are summarized from unstable, neutral, and stable atmospheric stability conditions. The turbulence spectrum and spatial coherence with separations in all directions are derived.

Based on the defined three turbulence cases, we analyzed the coherence between the rotor-effective wind speed and the one estimated by lidar. The NREL 5.0 MW reference wind turbine and a four-beam pulsed lidar system are taken into consideration. It is found that some differences appear between the results of the Mann model and that of the Kaimal model. The coherence using the Mann model is generally higher in all atmospheric stability classes than the coherence using the Kaimal model. We further analyzed the optimal transfer function, which is important to design a filter that removes the uncorrelated content in the lidar-estimated rotor-effective wind speed signal for lidar-assisted control. For most of the above rated wind speeds, the analysis revealed that the difference for the transfer function between using different turbulence models or different stability classes is not very significant. This also means a simple linear filter design for lidar-assisted control is sufficient for various atmospheric stability conditions. However, for wind speed above $20 \mathrm{~ms}^{-1}$, the cutoff frequency of unstable condition is about 0.02 Hz higher than that in the neutral stability. The non-ideal filtering should be further analyzed, which is cased by using the cutoff frequency derived from neutral stability for unstable stability. Also, the conclusions in this paragraph may not be applied to turbines of other sizes and lidars with other trajectories. The analysis of coherence and transfer function study can be extended for larger rotor turbines and other lidars with different trajectories.

To further analyse the impact of atmospheric stability for lidar-assisted control, a reference lidar-assisted control package is developed and used in this work. The lidar-assisted control package includes several DLL modules written in FORTRAN: 1) a wrapper DLL that calls all sub-DLLs sequentially, 2) the lidar data processing DLL that estimates the REWS and records the leading time of the REWS, 3) a feedforward pitch module that filters the REWS and activates the feedforward rate at the correct time, 4) a modified reference FB controller (ROSCO) which can receive feedforward command.

The benefits of lidar-assisted control are evaluated using both the Mann model and Kaimal model-based 4D turbulence. The simulations are performed for the mean wind speed level from $12 \mathrm{~ms}^{-1}$ to $24 \mathrm{~ms}^{-1}$, using the NREL 5.0 MW reference wind turbine and a four-beam lidar system. For the results with the Mann model, using lidar-assisted control reduces the variations in rotor speed, blade pitch rate, and electrical power significantly. Among the three investigated stability classes and above the mean wind speed of $16 \mathrm{~ms}^{-1}$, the load reductions for the tower bending moment, blade root bending moment, and low-speed shaft torque are observed to be approximately 3.0% to 16.7%, 1.5% to 6.0%, and 1.7% to 5.0%, respectively. The greatest potential of lidar-assisted control in load reduction is found in the tower base loads and the benefits are found to vary by turbulence spectral properties and mean wind speeds. For the results of the Kaimal model, using lidar-assisted control also reduces the variation in rotor speed, blade pitch rate, and electrical power clearly. The load reduction of the tower bending moment is found in all stability classes for wind speed above $16 \mathrm{~ms}^{-1}$ and it varies from 3.6% to 13.4%. The load reduction for the blade root bending moment is between 1.6% to 4.5% and for the low-speed shaft torque between 1.6% to 2.5%. Besides, with the help of lidar-assisted control, for both turbulence models, the standard deviation of pitch rate (speed) can be reduced (up to 6%,) for most of the mean wind speed range (below $20 \mathrm{~ms}^{-1}$) and for all stability classes. The pitch rate standard deviation reduction can bring potential load alleviation for the pitch bearings and gears. Overall, we found the benefits of lidar-assisted control by the Kaimal model are slightly different from the results obtained using the Mann model. The benefits of lidar-assisted control simulated using the Mann model is slightly better than that using the Kamal model, which can be caused by differences in the turbulence spatial coherence between two models. The lidar preview quality modeled using the Mann

model is generally superior to that modeled using the Kaimal model. For both turbulence models, there are clear trends that the benefits of lidar-assisted control in load reduction is the highest in unstable stability, medium in neutral stability, and lowest in a stable atmosphere.

With this work, we show that the mean wind speed, the turbulence spectrum, coherence, and the used turbulence models all have certain impacts on the results of evaluating lidar-assisted control. In this paper, the same turbulence intensity level is assumed for different atmospheric conditions. However, in reality, the turbulence intensity depends on the stability conditions of the atmosphere. In the future, we recommended assessing the benefits of lidar-assisted control depending on site-specific turbulence characteristics and statistics. Also, it is necessary to consider the uncertainties in turbulence models when performing

load analysis using aeroelastic simulations.

*Code availability.* The OpenFASTv3.0 version with a lidar simulator integrated can be accessed via: https://github.com/fengguoFUAS/OpenFAST3.0_LidSim or https://github.com/MSCA-LIKE/OpenFAST3.0_Lidarsim

The 4D Mann turbulence generator can be found by: https://github.com/fengguoFUAS/4D-Mann-Turbulence-Generator or https://github.com/MSCA-LIKE/4D-Mann-Turbulence-Generator

The open-access tool *evoTurb* has been published on Github: https://github.com/SWE-UniStuttgart/evoTurb.

The source codes of the wrapper DLL, baseline lidar data processing DLL, pitch feedforward DLL, and the modified ROSCO DLL are all available from https://github.com/fengguoFUAS/Baseline-Lidar-assisted-Controller or https://github.com/MSCA-LIKE/Baseline-Lidar-assisted-Controll

*Author contributions.* FG conceived the concept, performed the simulations and prepared the manuscript. DS supported to verify the simulations, provided general guidance and reviewed the paper. PWC provided suggestions, revised and reviewed the paper.

*Competing interests.* The authors declare that they have no conflict of interest.

*Acknowledgements.* This research received financial supports from the European Union's Horizon 2020 research and innovation program under the Marie Skłodowska-Curie grant agreement No. 858358 (LIKE – Lidar Knowledge Europe).

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
