# Peer review of "Evaluation of lidar-assisted wind turbine control under various turbulence characteristics"

_Wind Energy Science, 2022_

## Author Comment (AC1)

**Authors' response:**
**Evaluation of lidar-assisted wind turbine control under various turbulence characteristics**

Feng Guo*, David Schlipf and Po Wen Cheng

24.08.2022

First of all, we would like to sincerely thank all the reviewers for taking their valuable time to read our manuscript and provide constructive comments. We have carefully read and considered all the comments in detail and have revised our paper accordingly. Undoubtedly, the reviewers' comments have played a crucial role in improving the quality of our paper.

Please find below our response to the reviewers' comments. The reviewers' comments are repeated in black text, our response should be given in blue text, and if necessary, the corresponding corrections are provided in red text.

An important change from the first submitted version is that we updated the simulations using the newest ROSCO version (2.6.0) in the revised version. One of the most important update is that the pitch actuator model is included in the new version (previously not included in the 2.4.0). We think it is important to include a pitch actuator in realistic simulations for lidar-assisted control, since it limits the control performance of a feedback controller, while a feedforward controller can easily compensate the delay. There are small differences in the simulation statistics caused by this update.

Besides, we added a github link in the code availability section which includes the wrapper DLL and a baseline lidar-assisted controller.

**Response to comments of Anonymous Referee #1**

**Overall comments**

It is an interesting research work to evaluated the lidar-assisted wind turbine control under various turbulence characteristics using a four-beam liar and the NREL 5.0 MW reference turbine, which could be beneficial to the Lidar-assisted wind turbine control community. The paper is well organized and written. I recommend to accept the paper after considering and modifications are made.

We would like to thank the referee for the interest in reviewing this research and the positive feedback on the manuscript.

**Grammar and typos**

Line 24: missing white space between control ... (CPFF), please also cross-check in the whole content of the paper, this appears in many places. Thank you, this has been corrected in the revised version.

LIne 37: averaging → average. Thank you, this has been corrected in the revised version.

Line 55: ... (Schlipf, 2015) uses ... change to ... Schlipf uses ...Thank you, this has been corrected in the revised version.

LIne 81: "... designing ..." → design. Thank you, this has been corrected in the revised version.

Line 82: present → presents. Thank you, this has been corrected in the revised version.

LIne 90: structure → structures. Thank you, this has been corrected in the revised version.

line 106 a → an. Thank you, this has been corrected in the revised version.

line 129: $yz$ plane → $yz$-plane. Thank you, this has been corrected in the revised version.

line 175: "... Simley and Pao Simley and Pao (2015) ..." I guess the bracket around the cited reference is missing. Thank you, the citation here is wrong because the authors' names are repetitive. This has been corrected in the revised version.

line 176: what is Ih? it should be "In" right? Thank you, this is an typo. "In" is corrent. This has been corrected in the revised version.

line 190: Never use a symbol, e.g., "L" to start a sentence. Thanks for the suggestion, this has been corrected in the revised version.

line 276: propagates → propagate. Thank you, this has been corrected in the revised version.

line 277: try to replace semi-column with comma when seperating the cited references. This applies to all the context of your paper. Please cross check all of them. Thanks for the suggestion, all the semi-column has been replaced by comma in the revised version.

line 305: is → are. Thank you, this has been corrected in the revised version.

line 308: is → are. Thank you, this has been corrected in the revised version.

line 326: frequency → frequencies, are → is. cutoff → cut-off. Thank you, except for the term "cutoff", others has been corrected in the revised version. We kept using "cutoff" because this is more commonly used in literature. We also checked the paper and only "cutoff" is used in the revision.

line 327: that → those. Thank you, this has been corrected in the revised version.

line 340: delete "alone". Thank you, this has been corrected in the revised version. We know rewrite it as: "To make each controller as modularized as possible..."

line 395: contributed → affected? Thank you for pointing this out, we have updated the manuscript using your suggestion.

line 400: are → is Thank you, this has been corrected in the revised version.

line 477: by → in? Thank you, this has been revised as "However, the spectra estimated from simulated time series using the Mann model and the Kaimal model generally have similar shapes."

line 517: "Introduction the FF pitch ..." → "Introducing the FF pitch ..." or "Introduction of the FF pitch ..." Thank you for pointing this out, we use "introducing" now.

line 557: "... two turbulence ..." → "... two turbulence model ..." Thank you, this has been corrected in the revised version.

line 558: "... provided ..." → "... suggested ..." Thank you for pointing this out. We kept using "provided" to be in accordance with the previous text. The reason we did not use "suggested" is because Mann model is "recommended" by the IEC-61400:2019 standard but Kaimal model is not specified as "recommended". So writing that both models are "suggested" might be misleading.

line 570 - 571: " ... We further analyzed the transfer function, which is important for designing a filter, which removes uncorrelated content in the signal for lidar-assisted control." Please consider to rewrite this sentence. Thanks, this sentence is written as: "We further analyzed the optimal transfer function, which is important to design a filter that removes the uncorrelated content in the lidar-estimated rotor effective wind speed signal for lidar-assisted control."

line 593: "Overall, with this work ..." → "... with this work ..." Thank you, this has been corrected in the revised version.

**General comments**

1. line 43 to 45: the author states: "... two turbulence models are commonly used ...," But later 3 models are mentioned in the following sentence "... they are the Mann uniform shear model Mann (1994) and the Kaimal spectra Kaimal et al. (1972) and exponential coherence model (hereafter referred as to Mann model and Kaimal model respectively) ..."
Thank you for pointing this out. Indeed, the previous sentence is not very clear. Actually there are two models. We now rewrite the sentence as: "According to the IEC standard, two turbulence models are commonly used for wind turbine design as provided by the IEC 61400-1:2019 standard, one is the Mann (1994) uniform shear model and another one is the Kaimal spectra (1972) combined with exponential coherence model (hereafter referred as to Mann model and Kaimal model respectively)."

2. line 68 to 69: "... The length scale can have an impact on the power spectrum and turbulence spatial coherence." could you show an example to demonstrate this? Actually, this is discussed in Section 2.4. We added a bracket after the sentence to guide the readers to the discussion. (Now line 74)

3. Line 91: suggest to use vector notation for '$x$', e.g., $\vec{x}$
Thanks for the suggestion. However, we use the bold character ($\boldsymbol{x}$) to be in accordance with previous literature that are highly linked to this paper, such as: Mirzaei and Mann (2016), Held and Mann (2019) and Guo et al. (2022a).

4. line 100: what is $\Phi_{ij}(k)$ in Equation 3?
Thank you for pointing this out. We have added the definition of $\Phi_{ij}(k)$ in 106-112.

5. Line 162: Please double check the equation 15, the symbol $F_1 1(k_1)$ is wrongly typed in Latex.

Thanks for the careful review. It should be $F_{11}(k_1)$ and it has been corrected.

6. Sometimes, "evoturb" is used in the context, sometimes "evoTurb" is used, please unify them

Thank you for pointing this out. "evoTurb" is now used throughout the paper.

7. line 190 - 194: This description is redundate, because this has been mentioned in the introduction section

Thank you for pointing this out. We have removed these repetitive sentences.

8. line 210 - 211: "... Except for a relatively larger error for the $v$ component auto-spectrum under very unstable stability, the rest fittings show very good agreements. ...", I don't see this conclusion in Figure 2, Please double check this statement.

Thank you for pointing this out. It is a typo. It should be $u$ component not $v$ component. This is corrected in the revised version.

9. line 234 - 236: "... we summarize the lidar wind preview quality for the investigated four-beam lidar and the NREL 5.0MW reference turbine under different atmospheric stability classes. ..." what do you want to express? Maybe the author wants to express "the lidar wind preview quality for the NREL 5MW reference turbine under different atmospheric stability classes"?

Thank you for suggestion. The sentence is written as: "In the end, we summarize the wind preview quality of the investigated four-beam lidar for the NREL 5.0MW reference turbine under different atmospheric stability classes."

10. line 237: section 3.1, the procedure of calculating the "Turbine-estimated rotor effective wind speed" is missing. How do you get "$u_1(x)$"? by EKF estimator or other method?

Thank you for question. We should not use "turbine- estimated rotor effective wind speed" here, because here it is just how we define the "rotor-effective wind". As explained in Line 250, it is simply the averaged longitudinal components over the rotor-swept area. We have removed "turbine-estimated" to "turbine-based" in the revised paper.

11. line 309 - 310: "... The coherence in the unstable case is especially lower using the Kaimal model, which can be caused by the direct product method ...". Do you have any reference to support this statement?

The investigation by Simley (2015) using LES is the reference to support this statement in the following sentence. We made it clear by linking both sentences by a ":".

12. line 315: Please consider to re-formulate this sentence. "... If a filter with the gain GRL(f) turns out to be an optimal Wiener filter (Simley and Pao, 2013; Wiener et al., 1964), which results in minimal output variance for a multi-inputs multi-outputs system. ...". This sentence does mean anything.

Thanks for pointing this out, this sentence is now rewritten as: "If a filter is designed to have a gain of GRL(f), it turns out to be an optimal Wiener filter (Simley and Pao (2013), Wiener et al. (1964)), which results in minimal output variance for a multi-inputs multi-outputs system."

13. line 330 - 334: what about the cut-off frequency for different mean wind speed other than 16 m/s? The author needs to specify this.

Thanks for pointing this out, we have removed Table 5 and added Figure 5, which shows the cutoff frequencies as a function of the mean wind speed. We also added the discussions around the line 340 as: "The cutoff frequencies as a function of mean wind speed are calculated by fitting the GRL and are shown in Figure 5. Actually, both turbulence models indicate that the cutoff frequencies depend on the mean wind speed linearly. Therefore, the cutoff frequency of the filter can be scheduled based on this linearity."

14. line 374: equation 36, why the derivative of steady-state pitch angle is calculated with respect to Turbine estimated Rotor Effective Wind Speed ($u_{\mathrm{RR}}$)? and multiplied with ($u_{\mathrm{LLf}}$)) makes the equation mathmatically not exactly correct. What about using the Lidar estimated REWS when evaluating the derivative of steady-state pitch angle?

Thanks for pointing this out. Indeed, using $u_{\mathrm{RR}}$ here is not rigorous. We have changed from $u_{\mathrm{RR}}$ to $u_{\mathrm{ss}}$. Here, $u_{\mathrm{ss}}$ is the steady-state wind speed. We also added one sentence to specify how the steady-state pitch curve can be obtained.

15. line 408: "... 4096×11×64×64 grid points, corresponding to the time, and the $x$, $y$ and $z$ directions ...". This means to me only 11 grid points in the $x$ direction? (I suppose x- direction is the u-component direction). This seems to me too less grid points

Thanks for pointing this out. Indeed, 11 is the number of discrete points in the $x$ direction. We only discrete in the lidar measured $yz$-planes and the rotor plane. Because there are 10 lidar measurement planes and 1 rotor plane. There are 11 planes in total. Actually, lidar has a probe volume and in principle there should be more points in the $x$ direction. However, the study by Chen et al. (2022) shows that applying Taylor's frozen theory within the probe volume does not affect the lidar wind preview quality, as we discussed in line 463-467. Thus, we did not further discretize in the $x$ direction to save computer memory for the simulation.

16. In Figure 7, the time series of generator power should not have such kind of relative large oscillation because the author has mentioned that the constant Power mode (see line 367) is used in the simulation for above rated wind speed and 16 m/s

mean wind speed should well above rated and has less probability to be at below rated wind. Could you please explain this in your paper?

Thanks for pointing this out. Currently, ROSCO (v2.6.0) controller uses the low-pass-filtered generator speed to calculate the torque command by the formula: *generator torque = rated electrical power / generator efficiency / low-pass-filtered generator speed.* The main reason to use the filtered speed is to avoid response to motions with high natural frequencies, e.g. the shaft natural frequency and the measurement noise. If we do not consider the fact that turbine might have a short interval to go below rated operation during a wind speed trough. The formula above ensures that the electrical power is constant if the electrical power is calculated using the filtered generator speed. However, the actual electrical power (generator side not grid side) is determined by the non-filtered generator speed and the torque by: *electrical power = non-filtered generator speed \* generator torque \* generator efficiency = non-filtered generator speed / low-pass-filtered generator speed \* rated electrical power.* Therefore, the fluctuation of electrical power is contributed by the difference between the low-pass-filtered and non-filtered generator speeds. We have added relevant discussion in Section 4.3 and Section 5.2.1.

17. The followed up comments is as the follows: line 450 - 452: "... Lastly, the generator power is shown in panel (g) where much less power fluctuation is observed in FFFB control. Because the power fluctuation is highly coupled with the rotor speed fluctuation, the less fluctuating power can be expected from the less rotor speed fluctuation in FFFB control." This statement is not correct. As it was mentioned before, the constant power mode is used, what fluctatuated should be the generator torque and coupled to the rotor speed variations.

Thanks for your comment. Please see response 16 for the reason of power fluctuations. In Section 5.2.1, we actually want to point out that FFFB control can reduce the rotor speed fluctuation and the reduction in rotor speed is linked to the power fluctuations (by comparing Panel (b) and (g) in Figure 8). We agree with the reviewer that if constant power mode is applied using the non-filtered generator speed, then the power will be truly constant and the fluctuation will be the generator torque which is coupled to the rotor speed.

18. line 505 - 509: The statement is fair. It could be better to add some suggestions on how to solve this issue.

Thanks for pointing this out. We believe this can be solved by activating the feedforward loop smoothly only when the REWS is above a certain threshold or adjusting the weights of feedforward and feedback pitch commands. We have added the solutions to the second paragraph of Section 5.2.3

19. line 512 - 513: "... In the stable stability, the reduction is better at $14\,\mathrm{ms}^{-1}$, where the value is close to 4%, and it drops to 2% by higher wind speeds.". Does this mean for the stable atmosphere case, the probability of the wind speed lying in the transition range betweem below rated and above rate is lower than that of the

unstable atmosphere case? Adding a probability exceedance plot should help the discussion better.

Thanks for pointing this out. First, it should be $16\,\mathrm{ms}^{-1}$, as corrected by Referee 2. Here we want to point out that the LAC benefits reaches a peak value at a relatively lower wind speed for the stable condition. Firstly, for different stability conditions but same mean wind speed, it can be seen that the LAC benefits for the blade load is highest in unstable, medium in neutral, and lowest in stable. The reason could be the different length scales. The turbulence length scale is lower under stable condition which means the peak of turbulence spectrum appear at a higher wavenumber/frequency (base on the conversion $f = k_1 U_{\mathrm{ref}}/2\pi$). The blade load is mainly excited by frequency above 0.1 Hz (due to 1p 3p...), if the spectrum has a higher peak frequency, the load will be more dominated by the higher frequency parts. Then the LAC benefits becomes less because it mainly reduces loads below 0.1 Hz (for the lidar we used). Based on the discussion above, a higher mean wind speed shifts the spectral peak frequency to even higher frequency, thus the LAC benefits becomes less for the stable condition. For unstable and neutral cases, the spectrum peak frequency is naturally lower than that in the stable condition, thus the LAC benefits does not decrease as fast as that in the stable condition. We have added the discussion above to the end of Section 5.2.3.

20. The discussion between line 522 and line 524 should be explained. Please see the comments number 16 and 17.

Thanks for pointing this out. Please see the response on 16, 17 and 19. In the revised paper, we added more discussion based on the response of 19 to the end of section 5.

21. line 569 - 570: "... The coherence using the Mann model is generally higher in all atmospheric stability classes than the coherence using the Kaimal model. ...". For larger turbine, e.g., DTU10MW turbine, the coherence using the Mann model is generally much lower than the one using Kaimal model. The author needs to justify this in the context of his paper.

Thanks for the comment. In the end of this paragraph, we pointed out that the conclusions here may not be applied to turbines of other sizes and lidars with other trajectories. We also specify that the coherence and transfer function study can be extended for other turbine-lidar combinations.

**Response to comments of Anonymous Referee #2**

**General comments**

This is an interesting manuscript that uses stochastic wind fields and aeroelastic simulations to examine the effectiveness of lidar-assisted control in reducing loads, as well as the lidar measurement coherence, for three different stability classes using both the Mann and Kaimal turbulence models. This is an important topic because lidar-assisted control is typically only evaluated using the default Mann and Kaimal turbulence conditions, based on neutral stability. Evaluating lidar-assisted control in different conditions more accurately indicates how well the control strategy will work in the variety of conditions encountered by turbines during their lifetime. The authors provide a detailed overview of the turbulence models and simulation process and show relevant metrics when presenting the results. However, I have one major technical comment on the manuscript as well as many smaller comments that I believe should be addressed.

We would like to thank the referee for the interest in reviewing this research and giving feedbacks on the manuscript. We have carefully considered each comment, and please see the detailed responses below.

When simulating wind fields for all three stability classes, the same turbulence intensity is used for all cases (IEC class 1A). But in reality, stable atmospheric conditions will typically have much lower turbulence levels than unstable conditions, with neutral being somewhere in between. Therefore, the conditions being simulated likely don't represent stable, neutral, and unstable conditions very well. Would you be able to include more realistic TI values for each stability? Or can you discuss why you are using the same TI for each stability class?

Thanks for the your comments. We definitely agree with the comment that TI is generally high in unstable stability, moderate in neutral stability, and low in stable stability. We have thought about using different TI values for different atmospheric conditions in the beginning. However, when using different TI levels, it is hard to compare the load by different atmospheric conditions. Because both the turbulence length scale and TI changes by atmospheric stability conditions. In figure 11 and figure 12, it is easier to distinguish that the difference in load is mainly caused by the different turbulence length scales. To clarify this to the reader, we have added some discussions after line 192 and pointed out that the analysis considering different TI values can be made in future studies in the last paragraph of **Conclusions**.

The added content after line 192 is: Actually, the turbulence intensity is related to the atmospheric conditions. Usually, TI is generally high in unstable stability, moderate in neutral stability, and low in stable stability (Peña et al., 2017). In this work, we emphasize to analyze the impact of turbulence length scale on turbine loads and LAC benefits. Therefore, the same TI level is assumed for the three stability classes. This assumption tends to be less realistic but it helps to make loads more comparable in the analysis in Section 5.2.
The added content in the last paragraph is: "In this paper, the same turbulence intensity level is assumed for different atmospheric conditions, while in reality, the turbulence intensity depends on the stability conditions of the atmosphere."

Further, how accurate is the wind evolution model in the extended Mann model when using the unrealistic TI values for some stability classes? I assume it was developed using field measurements, but how well do these field measurements represent the class 1A turbulence simulated here for each stability class? Thanks for the your comments. The extended Mann model (space-time tensor) is developed using field measurement. Because the space-time tensor assumes stationary process in time, the turbulence intensity is not affected by the wind evolution. Actually the wind evolution parameter is determined by the parameter "$\gamma$". So, based on specific situations, one can adjust $\gamma$ and $\alpha\varepsilon^{2/3}$ independently to reach a target turbulence intensity and evolution level.

Another non-technical general comment is that there are many places in the manuscript where sentences are broken into two sentence fragments. For example, line 192: "It is clear that a larger coherent eddy structure... While the eddy structure is much smaller...", line 211: "It can be seen that the turbulence... While the variation in the anisotropy...", line 402: "To include the turbulence evolution... Four-dimensional stochastic turbulence..." I would suggest reviewing the manuscript and combining sentence fragments like these into single sentences. Thanks for the your comments. We have reviewed the paper and modified the fragmentary sentences.

**Specific comments**
1. Introduction: Much of the paper compares lidar measurement coherence between the Mann and Kaimal models. Since there has been some previous work in this area (e.g., Dong et al. (2021)), it would be helpful to discuss how this research compares to the previous work.
Thanks for the your comments. We added discussions in line 50 to make comparisons with the previous work.

2. Line 34: "The lidar measurements can be contaminated by lateral and vertical wind speed components": to understand why lateral and vertical wind speed components "contaminate" the lidar measurements, it would be helpful to explain what you are trying to estimate (i.e., how do you define the REWS you are trying to estimate. The rotor average of the longitudinal component?)
Thanks for the your comments. We added contents to clearify this. The added content is: "The turbine's aerodynamic performance is mainly driven by the $u$ component, and lidar is expected to measure the $u$ component for control purposes."

3. Line 54: Can you provided a reference for ROSCO?

Thanks for the your comments. We added the reference.

4. Fig. 1 caption: Can you provide a reference for how the length scales were chosen for each stability class?
Thanks for the your comments. We added the references.

5. Line 70: "Because the turbulence spectrum peaks..." This is an incomplete sentence.
Thanks for the your comments. We have corrected this sentence in the revised version as: "Further, the spectrum and coherence can have potential impacts not only on the lidar measurement coherence but also on the turbine loads because the turbulence spectrum peaks can distribute at different frequency ranges, and different frequencies can produce different excitations for the turbine structure motion."

6. Section 2.2: Are you assuming zero spatial coherence for the lateral and vertical velocity components? It would be helpful to discuss this here.
Thanks for the your comments. We actually ignored the $yz$ plane coherence for lateral and vertical velocity components. We added discussion in the end of Section 2.2 as "The $yz$ plane coherence for the $v$ and $w$ components are not given by the IEC 61400-1:2019 and they are ignored in this work."

7. Section 2.3.1: The extended Mann model with evolution clearly shows a dependence on length scale (e.g., Eq. 14). Can you discuss how other wind conditions, such as turbulence intensity, affect the coherence? For example, in Simley and Pao (2015) there is a strong relationship between TI and coherence, but it isn't clear how this is captured in the extended Mann model.
Thanks for the your comments. The extended Mann model (space-time tensor) assumes stationary process in time, the turbulence intensity is not affected by the wind evolution. Actually the wind evolution parameter is determined by the parameter "$\gamma$". So, based on specific situations, one can adjust $\gamma$ and $\alpha\varepsilon^{2/3}$ independently to reach a target turbulence intensity and evolution level.

8. Line 186: "impact on filter design for LAC": I would suggest explaining what filter you are referring to here.
Thanks for the your comments. We added some contents to explain the functionality of the filter. i.e. The filter is necessary to filter out the uncorrelated frequencies in the REWS estimated by lidar, as will be discussed later in Section 3.

9. Eq. 20: Why is the real number operator needed here? By definition, won't the coherence be a positive real number? Otherwise, can you explain how $\text{coh}_{11}$ can contain imaginary components?
Thanks for the your comments. Indeed, the magnitude squared coherence is real and positive. In terms of the least square fitting in Equation (21) (previously 20), we are fitting the co-coherence. We have corrected the equation now and added

Equation (10) to explain the definition of co-coherence.

10. Line 212: "while the variation in the anisotropy Gamma does not show a clear trend towards the atmospherically." Based on Table 1, there is a clear trend between Gamma and stability. Are the values for Gamma and length scale in Table 1 switched perhaps?

Thanks for the question. We have checked the values of Gamma and the sequence is correct. Based on the study by Peña et al. (2017) and Peña (2019), (the measurements from Danish sites), $\Gamma$ also show unclrear trend by stability.

11. Line 219: "we use three sets of gamma = 200, 400, and 600 s" Why did you choose these three values?

Thanks for the question. We have added the reason as: "The reason for choosing these values for $\gamma$ is that they result in coherence close to observations in existing literature, as will be discussed later".

12. Line 222: "which is the median separation for a commercial lidar measuring in front of the turbine" Can you provide a reference or list some examples of commercial lidars and their measurement ranges?

Thanks for the suggestion. We have added two references that describes the recent commercial lidars.

13. Line 229-231: It is unclear what you mean by "rarely large $a_x$" and why this suggests you should use gamma = 600 s for the unstable case. More generally, can you discuss in more detail why you chose 600 s to represent the unstable condition (e.g., why not 500 s or 800 s)? Further, can you discuss how accurate the selected gamma values are for the class 1A turbulence intensity used in the simulations? And how would gamma change for different TI values? (e.g., Simley and Pao (2015) observed a strong relationship between TI and coherence).

Thanks for the question. We use the term "rarely" according to the probability study by Chen et al. (2020), but we did not write it clearly previously. We chose 600 s because it gives higher $a_x$ in the unstable condition than the neutral and stable conditions (in accordance with the LES-based observation by Simley and Pao (2015)). Overall, 600 s is chosen because it gives a reasonable $a_x$ value in terms of probability and relative difference with $a_x$ from other stability. Now we have modified the sentence to be more clear. Regarding the second question, as explained in the general comment, we have not consider the variation in the TI to emphasize the study on the changes in turbulence length scale. The $\gamma$ value is independent from the turbulence intensity in the space-time tensor. Also, since the turbulence intensity is adjusted by the parameter $\alpha\varepsilon^{2/3}$, which is just a proportional gain. The changes in the $\alpha\varepsilon^{2/3}$ will not affect the coherence of velocity components or lidar measurement. In reality, one can design simulations using different $\gamma$ values to reach the target longitudinal coherence.

14. Line 257: "azimuth angle phi and elevation angle beta" The math is hard to follow in this section without understanding how the azimuth and elevation angles are defined. Can you define these angles or show them in a figure?
Thanks for the suggestion. We added the definitions in Figure 3.

15. Eq. 27-32: Shouldn't the angles phi and beta be a function of the lidar beam and therefore depend on the index "i"?
Thanks for the suggestion. In our previous study, most of the lidar systems have identical value of $\cos\beta\cos\phi$, so that we did not use index for simplicity. However, this is limited to specified lidar system. To make the formula more general to any lidar trajectory, we have adopted the reviewser's suggestion and modified these equations.

16. Eq. 31: I think there should be the imaginary number "i" in front of "$k_1\Delta x_i$". Also, as written, because $\Delta x_i$ equals $x_i$, it seems that $S_{\mathrm{RL}}(k_1)$ won't contain the phase delays between the measurement points and the rotor because the $k_1$ dependence of the exponent simplifies to $\exp(\mathrm{i}(k_1 * x_1 - k_1 * x_1)) = 1$. Should $\Delta x_i$ in the equation simply be replaced by $x_{\mathrm{R}}$ to model the correct phase delay?
Thanks for the careful review. The reviewer is correct, the imaginary number "i" should be included in front of "$k_1\Delta x_i$". This has now be added. As for the second question, we added the detailed derivation below: The turbulence field can be represented by the Fourier transform as (Mirzaei and Mann, 2016):

$$\boldsymbol{u}(\boldsymbol{x},t) = \int \hat{\boldsymbol{u}}(\boldsymbol{k},t)\exp(\mathrm{i}\boldsymbol{k}\cdot\boldsymbol{x})\mathrm{d}\boldsymbol{k}, \tag{1}$$

The lidar LOS measurement can be approximated by

$$
\begin{aligned}
v_{\mathrm{los}}(\boldsymbol{x},t) &= \int_{-\infty}^{\infty} \varphi(s)\boldsymbol{n}\cdot\boldsymbol{u}(s\boldsymbol{n}+\boldsymbol{x},t)\mathrm{d}s \\
&= \int \boldsymbol{n}\cdot\hat{\boldsymbol{u}}(\boldsymbol{k},t)\exp(\mathrm{i}\boldsymbol{k}\cdot\boldsymbol{x})\int_{-\infty}^{\infty}\varphi(s)\exp(\mathrm{i}\boldsymbol{k}\cdot\boldsymbol{n}s)\mathrm{d}s\mathrm{d}\boldsymbol{k} \\
&= \int \boldsymbol{n}\cdot\hat{\boldsymbol{u}}(\boldsymbol{k},t)\exp(\mathrm{i}\boldsymbol{k}\cdot\boldsymbol{x})\hat{\varphi}(\boldsymbol{k}\cdot\boldsymbol{n})\mathrm{d}\boldsymbol{k}.
\end{aligned}
\tag{2}
$$

The rotor effective wind speed is defined as (Mirzaei and Mann, 2016):

$$
\begin{aligned}
u_{\mathrm{RR}}(x_1,t_0) &= \frac{1}{\pi R^2}\int_D u(\boldsymbol{x},t_0)\mathrm{d}y\mathrm{d}z \\
&= \frac{1}{\pi R^2}\int_D\int \hat{u}(\boldsymbol{k},t_0)\exp(\mathrm{i}\boldsymbol{k}\cdot\boldsymbol{x})\mathrm{d}\boldsymbol{k}\mathrm{d}y\mathrm{d}z \\
&= \frac{1}{\pi R^2}\int \hat{u}(\boldsymbol{k},t_0)\exp(\mathrm{i}k_1 x_1)\int_D\exp(\mathrm{i}k_2 x_2+\mathrm{i}k_3 x_3)\mathrm{d}y\mathrm{d}z\mathrm{d}\boldsymbol{k} \\
&= \int \hat{u}(\boldsymbol{k},t_0)\exp(\mathrm{i}k_1 x_1)\frac{2J_1(\kappa R)}{\kappa R}\mathrm{d}\boldsymbol{k}
\end{aligned}
\tag{3}
$$

where $D$ denotes the integration over the rotor area defined by rotor radius $R$. For simplicity, we consider only one lidar measurement position, then the Fourier transform (towards $x_1$, non-unitary convention) of the lidar- and turbine- based rotor effective wind speed are

$$
\begin{aligned}
\hat{u}_{\mathrm{LL}}(k_1, t) &= \frac{1}{\cos\beta\cos\phi} \int_{-\infty}^{\infty} \int \boldsymbol{n} \cdot \hat{\boldsymbol{u}}(\boldsymbol{k}, t) \exp(\mathrm{i}\boldsymbol{k}\cdot\boldsymbol{x} - \mathrm{i}k_1 x_1) \hat{\varphi}(\boldsymbol{k}\cdot\boldsymbol{n}) \mathrm{d}\boldsymbol{k}\mathrm{d}x_1 \\
&= \frac{1}{\cos\beta\cos\phi} \int \boldsymbol{n} \cdot \hat{\boldsymbol{u}}(\boldsymbol{k}, t) \exp(\mathrm{i}k_2 x_2 + \mathrm{i}k_3 x_3) \hat{\varphi}(\boldsymbol{k}\cdot\boldsymbol{n}) \mathrm{d}k_2 \mathrm{d}k_3,
\end{aligned}
\tag{4}
$$

and

$$
\begin{aligned}
\hat{u}_{\mathrm{RR}}(k_1, t_0) &= \int_{-\infty}^{\infty} \int \hat{u}(\boldsymbol{k}, t_0) \exp(\mathrm{i}k_1 x_1 - \mathrm{i}k_1 x_1) \frac{2 J_1(\kappa R)}{\kappa R} \mathrm{d}\boldsymbol{k}\mathrm{d}x_1 \\
&= \int \hat{u}(\boldsymbol{k}, t_0) \frac{2 J_1(\kappa R)}{\kappa R} \mathrm{d}k_2 \mathrm{d}k_3.
\end{aligned}
\tag{5}
$$

Note the integration over $k_1$ is annihilated because the right side has to be a function of $k_1$. Then, the cross spectrum is simply

$$
\begin{aligned}
S_{\mathrm{RL}}(k_1) &= \hat{u}_{\mathrm{LL}}(k_1, t)\hat{u}_{\mathrm{RR}}^*(k_1, t_0) \\
&= \frac{1}{\cos\beta\cos\phi} \int \boldsymbol{n} \cdot \hat{\boldsymbol{u}}(\boldsymbol{k}, t) \exp(\mathrm{i}k_2 x_2 + \mathrm{i}k_3 x_3) \hat{\varphi}(\boldsymbol{k}\cdot\boldsymbol{n}) \mathrm{d}k_2 \mathrm{d}k_3 \\
&\quad \int \hat{u}(\boldsymbol{k}', t_0) \frac{2 J_1(\kappa R)}{\kappa R} \mathrm{d}k_2' \mathrm{d}k_3' \\
&= \frac{1}{\cos\beta\cos\phi} \sum_{j=1}^{3} \int n_j \Theta_{j1}(\boldsymbol{k}, t - t_0) \exp(\mathrm{i}k_2 x_2 + \mathrm{i}k_3 x_3) \hat{\varphi}(\boldsymbol{k}\cdot\boldsymbol{n}) \frac{2 J_1(\kappa R)}{\kappa R} \mathrm{d}k_2 \mathrm{d}k_3.
\end{aligned}
\tag{6}
$$

The equation above is identical to Equation (32) (single lidar measurement position case) in the manuscript. In the manuscript, we also changed from $k_1 \Delta x_i$ to $k_1 x_i$ because $k_1 \Delta x_i$ is not correct when $x_{\mathrm{R}} \neq 0$.

17. Line 288: "the $i$th lidar measurement position" Can you clarify whether the index "$i$" refers to the lidar measurement position (e.g., combination of beam and range gate) or just the lidar beam? Earlier on line 267, the index "$i$" was described as representing the beam number.
Thanks for the comment. We refer to the lidar measurement position here. We adjusted the content in line 267 and $i$ also means measurement position. To avoid misleading, we now uses $n_{\mathrm{L}}$, instead of $n_{\mathrm{b}}$ to denote the number of measurement positions. This makes the formula also applicable to pulsed lidars.

18. Line 302: "typical four-beam pulse lidar trajectory": Can you discuss why this is "typical"? Are there commercial examples you could reference?
Thanks for the comment. We added the names of the commercial examples.

19. Table 3: As mentioned in an earlier comment, the azimuth and elevation angles haven't been defined. Can you define these or show them in a figure?
Thanks for the suggestion. We added the definitions in Figure 3.

20. Line 330: What are the units of the cutoff frequency $10^-3$?
Thanks for the comment. We added the unit Hz.

21. Line 330: "This also indicates that the filter design is not sensitive to the change in turbulence parameters... and a constant filter design is robust." How does the filter design depend on the wind speed? Do the cutoff frequencies change?
Thanks for pointing this out, we have removed Table 5 and added Figure 5, which shows the cutoff frequencies as a function of the mean wind speed. We also added the discussions around the line 340 as: "The cutoff frequencies as a function of mean wind speed are calculated by fitting the GRL and are shown in Figure 5. Firstly, both turbulence models indicate that the cutoff frequencies depend on the mean wind speed linearly. Therefore, the cutoff frequency of the filter can be scheduled based on this linearity"

22. Table 4: Please specify the units of the frequencies
Thanks for the comment. We added the unit in the Caption.

23. Line 340: "to make each controller module as standard alone as possible," This sentence is a little confusing. What do you mean by "standard alone"?
Thank you for the comment, we actually want to say "modularzied". This has been corrected in the revised version. We know rewrite it as: "To make each controller as modularized as possible..."

24. Section 4.2: Do you model the time delay between measurement points due to the sequential scanning of the lidar in the simulations or assume that each point is measured at the same time?
Thanks for the comment. We considered the sequential scanning in the simulation and we added descriptions in Section 4.2.

25. Line 354: "the blockage effect" usually refers to the reduction in wind speeds upstream of a wind farm. Is this what you are referring to here? If not, I would suggest clarifying or using a different term.
Thanks for the comment. We now use "blade blockage effect" to make it more clear .

26. Line 367: "we have chosen the option of constant power mode": Can you explain this control mode for readers unfamiliar with the term?
Thanks for the comment. We added description as: "In terms of generator torque control in the above-rated operation, we have chosen the option of constant power mode in our simulations, with which the generator torque is set according to the filtered generator speed to keep the electrical power close to its rated value. The

generator torque ($M_\mathrm{g}$) is set according to the low-pass-filtered generator speed, the rated electrical power ($P_\mathrm{rated}$), and the generator efficiency ($\eta$) by $M_\mathrm{g} = P_\mathrm{rated}/\eta\Omega_\mathrm{gf}$."

27. Eq. 36: This equation is hard to understand. Wouldn't the feedforward pitch command simply be $\theta_\mathrm{ss}(u_\mathrm{RR})$ (i.e., the steady-state pitch angle as a function of wind speed)? Otherwise, please discuss why this equation is used and how it is derived.
Thanks for the comment. In our controller implementation, the feedforward pitch command is obtained by interpolating the steady-state pitch curve using the lidar estimated REWS. So it should be $\theta_\mathrm{FF} = \theta_\mathrm{ss}(u_\mathrm{LLf})$.

28. Line 385: What is the value of the actuator delay that is used?
Thanks for the comment. We added the description of the used pitch actuator and the corresponding delay (0.22 second). The delay is determined by the phase delay of the pitch actuator system model.

29. Eq. 39: This equation is also hard to understand. It seems like it is missing the actual time delay that you are trying to solve for. Also, should the 1-second lidar averaging delay be included here too? Further, is there enough lead time to account for the filter, pitch, and lidar averaging times for all cases analyzed?
Thanks for the comment. We adjusted the formula and added some descriptions about each time delay. We also added Figure 8 to show the leading time of the lidar measurement gate and the required leading time for pitch forward control. In the case that leading time of range gate 1 is insufficient (for high wind speed), then gate 2 to 10 are used to estimated REWS and provide pitch forward command. We also added analysis in Figure 5 to show that losing one measurement gate does not has an significant impact on the cutoff frequency.

30. Line 411: What is the mean flow field used for the Kaimal model-based wind fields? Is it the same as the power law shear mean flow field used for the Mann wind field?
Thanks for the comment. Yes, same mean profile is used for Kaimal model-based turbulence fields, we have added a sentence to clarify this.

31. Line 414: "The lengths in the y and z directions are both 150 m": It would be good to discuss why these lengths are smaller than for the Mann wind fields.
Thanks for the comment. We added discussion to explain the reason. It is because that there is no periodicity problem in the Kaimal-model based turbulence field. And the field size are enough to simulate the aerodynamic of the NREL 5.0 MW turbine.

32. Line 416: "hub height wind speed from 14 m/s to 24 m/s with a step of 2 m/s are considered." It would be helpful to include simulation results for 12 m/s because this is above rated for the NREL 5 MW turbine and lidar-assisted pitch

control would be active. Additionally, Table 4 lists the cutoff frequencies of the lidar filter for 16 m/s. How do the cutoff frequencies change for the different wind speeds simulated?

Thanks for the comment, we have added the simulation results for $12 \, \mathrm{ms}^{-1}$. A cutoff frequency as a function of the mean wind speed has also been added. For details, please see the 13th response to Referee 1.

33. Fig. 7: Since you are using the constant power control mode (where typically torque is controlled to maintain constant power regardless of generator speed), it is surprising to see such high power fluctuations. Can you discuss why this is the case?

Thanks for pointing this out. Please see comment 16 of first reviewer.

34. Line 454: "The spectra are averaged by different samples corresponding to the simulated results by different random seed numbers." This sentence is hard to understand.

Thanks for the comment, we have adjusted the sentence as: "The spectra are averaged over different samples. Each sample is obtained from the aeroelastic simulation result produced by a turbulence field, which is generated by a specific random seed number."

35. Line 461: "In the unstable case, the RWES spectrum does not reduce a lot compared to a single point u spectrum..." To illustrate this point, it would be helpful to include the single point spectra in Fig. 8.

Thanks for the comment, we added the $u$ component in the plot (now Figure 10).

36. Line 474: "...which can be summarized as higher spectra in the rotor motion by the Kaimal model than the Mann model." It would be easier to see this if you used the same y axis limits in Figs. 9 and 10.

Thanks for the suggestion, we have adjusted the scales.

37. Line 481: "The reduction in the blade root out-of-plane motion is not very observable from the plots..." But significant reduction between 0.02 and 0.2 Hz can also be observed.

Thanks for the comment. We have changed the description to ":There are slightly reductions in the blade root out-of-plane moment in the frequency range from 0.02 Hz to 0.1 Hz contributed by LAC. It can also be seen that the spectrum is mainly composited by the excitation at the 1P frequency." We describe the reduction as slightly, not as significant because the y scale is set to be logarithmic. In Figure 13 and 14, it can be seen that the reduction is below 5 percent.

38. Section 5.2.3: It would be nice to add some more discussion to this section, for example, providing some reasons why the load reduction from lidar-assisted control might be different for the different stability classes.

Thanks for the comment. We have added discussion at the end of this section as:

"For different stability conditions but the same mean wind speed, it can be seen that the LAC benefits for the load reduction are overall highest in unstable, medium in neutral, and lowest in stable. The reason could be the difference in turbulence length scales. The turbulence length scale is lower under stable condition which means the peak of turbulence spectrum appear at a higher wavenumber/frequency (based on the conversion $f = k_1 U_{\mathrm{ref}}/2\pi$). The turbine structural loads are mainly excited by frequency above 0.1 Hz, e.g. the tower natural frequency, the shaft natural frequency (above 1 Hz), the 1P and the 3P frequency. If the spectrum has a higher peak frequency, the loads will be more dominated by the higher frequency parts due to the higher excitation of the natural modes. Then the LAC benefits become less significant because it mainly reduces the loads below 0.1 Hz (for the lidar and turbine we used). When considering different mean wind speeds, the discussions above indicates that a higher mean wind speed shifts the spectral peak-frequency to be a higher value, thus the LAC benefits becomes less. For the stable condition, the spectral peak frequency is naturally high due to the smaller turbulence length scale, thus it is more sensitive to the changes in the mean wind speed. For unstable and neutral cases, the spectrum peak frequency is naturally lower than that in the stable condition, thus the LAC benefits does not decrease as fast as that in the stable condition."

39. Line 491: Can you provide a reference for "rain flow counting"?
Thanks for the comment. We have added the reference.

40. Line 495: "For rotor speed, pitch rate... the standard deviation... is calculated". What is the significance of the std. dev. of pitch rate? Is this a common metric for pitch actuator damage? Why would this be used instead of the std. dev. of pitch angle, or the average pitch rate, etc.?
Thanks for the comment. We use the standard deviation of pitch rate (speed) to assess the impact of different control methods on the pitch actuator (also used by Chen and Stol (2014) and Jones et al. (2018)) because pitch speed causes damping torque in the pitch gear and is related to the friction torque of the pitch bearing (see e.g., Shan (2017) and Stammler et al. (2018)). A more fluctuating pitch speed will causes higher fatigue cycles for the gear and the bearing. We added the explanations at the beginning of Section 5.2.3.

41. Line 522: "As for the electric power STD...": Again, why is there any significant power fluctuation, since the constant power control model is used?
Thanks for the comment. Please see the response to 33.

42. Line 524: Why is there a significant reduction in mean power at 14 m/s?
Thanks for your comment. Because of turbulent wind, there are duration's when wind speed goes below the rated 11.4m/s. The power production there cannot reach 5MW and it has an impact on the mean power. The single point turbulence and also the REWS by Mann model and Kaimal model follows Gaussian distribution. A lower mean wind speed has a higher probability to drop below rated. This is the

reason that the mean power does not reduce when the mean wind speed is above 16m/s. We now analyzed the reason for power drop at 14m/s in line 583.

43. Line 546: "the electricity productions are similar either using LAC or not..." Again, there is a significant drop in power at 14 m/s with LAC. What causes this? Thanks for your comment. The reason of the lower mean power at 14m/s has been explained in 42. We now analyzed the reason for power drop at 14m/s in line 582. However, we did not get the reviews opinion that the EP is reduced with LAC. In the plot, the right side $y$ axis is the relative reduction compared to FB-only control. If it is negative value, it means the FFFB gives higher value compared to FB-only control. So the EP is actually increased (very slightly) with LAC. However, since the increment is marginal, our conclusion is that LAC has marginal impact on the EP.

**Minor comments**

1. In many places throughout the manuscript, there are citations without parentheses, for example line 44: "Mann (1994)." If the reference is actively used as part of the sentence, it is ok to leave the parentheses out, such as lines 46-48. Otherwise, I suggest using parentheses, for example, as is done in line 25.
Thanks for the careful suggestion. We have went through the paper and corrected relative citations.

2. Eq. 15: Should the second "1" in $F_1 1$ be a written as a subscript as well? Thanks for the careful review. It should be $F_{11}(k_1)$ and it has been corrected.

3. Eq. 23: Can $dk_2 dk_3$ be replaced by the symbol used in Eq. 8? Thanks for the careful review. The symbol "$d\boldsymbol{k}_\perp$" is used now.

4. Line 315: "If a filter with the gain..." This sentence is hard to understand and appears to be incomplete. Thanks for pointing this out, this sentence is now rewritten as: "If a filter is designed to have a gain of GRL(f ), it turns out to be an optimal Wiener filter (Simley and Pao (2013), Wiener et al. (1964)), which results in minimal output variance for a multi-inputs multi-outputs system."

5. Line 321: "natural" → "neutral. Thanks for the careful review. It has been corrected.

6. Line 436: "Karmann" → "Kalman". Thanks for the careful review. It has been corrected.

7. Line 512: "14 m/s" → "16 m/s"? 8. Line 541: "16 m/s" → "18 m/s"? Thanks for the careful review. It has been corrected.

9. Line 627: The paper "Dong et al. 2021" has been published as a full paper,

so the reference should be updated. Thanks for the careful review. It has been updated.

---

## Author Response (AR2)

**Authors' response to Review 2 (2nd revision):**
**Evaluation of lidar-assisted wind turbine control under various turbulence characteristics**

Feng Guo*, David Schlipf and Po Wen Cheng

16.11.2022

We would like to sincerely thank all the reviewers for taking their valuable time to read our manuscript and provide constructive comments. Special thanks to Review 2 for your careful review of the paper and further constructive comments in the second round of review.

Please find below our response to the reviewer 2's comments. The reviewer's comments in the report are repeated in black text, our response should be given in blue text, and if necessary, the corresponding corrections are provided in red text.

**Response to comments of Anonymous Referee #2 (2nd round)**

**Overall comments**

The revised manuscript has been greatly improved. I do have some remaining comments that I feel should be addressed, though. The main comments are related to a) including more discussion about how accurate the wind evolution parameters (gamma) used in the analysis are for different TI values, since you are just using a single value of gamma to represent each stability class regardless of the TI, and b) the assumptions made in the lidar spectrum and lidar-REWS cross spectrum equations. Please see responses to the remaining comments below (note: the comment numbers are different than the original numbers). Several new, mostly minor, comments are provided afterwards as well.

We would like to thank the referee for the positive feedback after the first round of revision on the manuscript.

1. Reviewer comment 1: Another non-technical general comment is that there are many places in the manuscript where sentences are broken into two sentence fragments. For example, line 192: "It is clear that a larger coherent eddy structure. . . While the eddy structure is much smaller. . . ", line 211: "It can be seen that the turbulence. . . While the variation in the anisotropy. . . ", line 402: "To include the turbulence evolution. . . Four-dimensional stochastic turbulence. . . " I would suggest reviewing the manuscript and combining sentence fragments like these into single sentences.
Author response: Thanks for the your comments. We have reviewed the paper and modified the fragmentary sentences.
Reviewer comment 2: Many of these issues were resolved. However, there are still some incomplete sentences throughout the manuscript. For example, Figure 1 caption "Simulated using the 4D Mann. . . "; Ln 192: "To include the exponential longitudinal coherence model. . . "; Ln 357: "Apart from the case that all measurement gates. . . "; Ln 432: "Because the pitch curve has much higher. . . "; Ln. 503: "If we do not consider. . . "; Ln. 522: "Because the turbulence field has a. . . "
Thanks a lot for the reviewer's careful reading. We have modified the text to fix these issues. We also further examined the full text of the revised manuscript.

2. Reviewer comment 1: Section 2.3.1: The extended Mann model with evolution clearly shows a dependence on length scale (e.g., Eq. 14). Can you discuss how other wind conditions, such as turbulence intensity, affect the coherence? For example, in Simley and Pao (2015) there is a strong relationship between TI and coherence, but it isn't clear how this is captured in the extended Mann model. Author response: Thanks for the your comments. The extended Mann model (space-time

tensor) assumes stationary process in time, the turbulence intensity is not affected by the wind evolution. Actually the wind evolution parameter is determined by the parameter "$\gamma$". So, based on specific situations, one can adjust $\gamma$ and $\alpha\epsilon^{2/3}$ independently to reach a target turbulence intensity and evolution level. Reviewer comment 2: I understand that the turbulence intensity isn't affected by the wind evolution, but I am wondering how the wind evolution parameter gamma in Eq. 15 depends on the turbulence intensity. For example, Simley and Pao (2015) observed a strong relationship between turbulence intensity and the $a_x$ parameter (in Eq. 18). Although Davoust and von Terzi (2016) didn't observe as strong of a relationship, there may be some dependence of the gamma parameter on TI. See comment #5 also.

Thanks a lot for the reviewer's comments. We have added some text in the end of section to point out the current shortage of our assumption. And propose suggestions for future research. The added text are: In addition, it is worth mention that we do not consider the dependence of the turbulence evolution parameters on TI level. The selection of turbulence evolution parameters is based on relevant studies and typical values are chosen. As studied by Simley and Pao (2015), the TI values can be different for the same atmospheric stability, and the evolution parameters show some dependence on the TI values. In the future, a joint probabilistic study on the turbulence spectral parameters, TI levels, and evolution parameters is necessary for defining more realistic simulation scenarios for LAC.

3. Reviewer comment 1: Eq. 20: Why is the real number operator needed here? By definition, won't the coherence be a positive real number? Otherwise, can you explain how coh11 can contain imaginary components? Author response: Thanks for the your comments. Indeed, the magnitude squared coherence is real and positive. In terms of the least square fitting in Equation (21) (previously 20), we are fitting the co-coherence. We have corrected the equation now and added Equation (10) to explain the definition of co-coherence. Reviewer comment 2: Can you explain why you are fitting the co-coherence instead of the magnitude squared coherence? Thanks a lot for the reviewer's comments. We chose to fit the co-coherence because the exponential coherence model (used for Kaimal spectra) is a real function and it only has co-coherence. We have added explanations in Ln 220.

4. Reviewer comment 1: Line 219: "we use three sets of gamma = 200, 400, and 600 s" Why did you choose these three values? Author response: Thanks for the question. We have added the reason as: "The reason for choosing these values for $\gamma$ is that they result in coherence close to observations in existing literature, as will be discussed later". Reviewer comment 2: I have one minor comment, which is to be more specific about which section this will be discussed in "later". Thanks a lot for the reviewer's comments. We have added text to point out the explanation is given at the end of the section.

5. Reviewer comment 1: Line 229-231: It is unclear what you mean by "rarely

large $a_x$" and why this suggests you should use gamma = 600 s for the unstable case. More generally, can you discuss in more detail why you chose 600 s to represent the unstable condition (e.g., why not 500 s or 800 s)? Further, can you discuss how accurate the selected gamma values are for the class 1A turbulence intensity used in the simulations? And how would gamma change for different TI values? (e.g., Simley and Pao (2015) observed a strong relationship between TI and coherence). Author response: Thanks for the question. We use the term "rarely" according to the probability study by Chen et al. (2020), but we did not write it clearly previously. We chose 600 s because it gives higher ax in the unstable condition than the neutral and stable conditions (in accordance with the LES-based observation by Simley and Pao (2015)). Overall, 600 s is chosen because it gives a reasonable ax value in terms of probability and relative difference with $a_x$ from other stability. Now we have modified the sentence to be more clear. Regarding the second question, as explained in the general comment, we have not consider the variation in the TI to emphasize the study on the changes in turbulence length scale. The $\gamma$ value is independent from the turbulence intensity in the space-time tensor. Also, since the turbulence intensity is adjusted by the parameter $\alpha\epsilon^{2/3}$, which is just a proportional gain. The changes in the $\alpha\epsilon^{2/3}$ will not affect the coherence of velocity components or lidar measurement. In reality, one can design simulations using different $\gamma$ values to reach the target longitudinal coherence.

Reviewer comment 2: The added discussion helps clarify the choice of gamma = 600 s for the unstable case a lot. Regarding the comment about how accurate these values are for the class 1A turbulence intensity used, I understand that gamma is independent from TI in the model and you are free to use any combination of TI and gamma. But since gamma is an additional free variable, it has to be tuned, as discussed in the manuscript. It therefore could depend on TI (or other variables) in addition to stability. You chose three values of gamma for the three stability classes, but are these choices of gamma valid for all TI values within a certain stability class? It would be insightful if you could discuss how accurate you believe the choices of gamma are for the class 1A TI values you use in the paper and it would help to acknowledge that the three values selected may not be accurate for all TI values (including the class 1A TI used in the paper) if that is the case.

Thanks a lot for the reviewer's comments. We have added text in the end of section 2. We specify the realistic phenomenons that we did not consider and propose suggestions for further works. Please see Comment 2.

6. Reviewer comment 1: Eq. 31: I think there should be the imaginary number "i" in front of "$k_1 \Delta x_i$". Also, as written, because $\Delta x_i$ equals $x_i$, it seems that $SRL(k_1)$ won't contain the phase delays between the measurement points and the rotor because the $k_1$ dependence of the exponent simplifies to $\exp(i(k_1 \cdot x_1 - k_1 \cdot x_1) = 1$. Should $\Delta x_i$ in the equation simply be replaced by $x_R$ to model the correct phase delay? Author response: Thanks for the careful review. The reviewer is correct, the imaginary number "i" should be included in front of "$k_1 \Delta x_i$". This has now be added. As for the second question, we added the detailed derivation below:...

Reviewer comment 2: Thanks for providing the derivations. I also see that the equation is nearly identical to the equation presented in Held and Mann, 2019. However, it still isn't clear why there doesn't appear to be any phase rotation for the $k_1$ frequency component due to the time shift between the lidar measurements and the REWS at the rotor plane in Eq. 32, assuming Taylor's hypothesis (i.e., $\exp(j \cdot k_1 \cdot \Delta_x)$), since the $k_1 x_i$ terms cancel out in the equation. Further, if the lidar-estimated REWS contains measurements at different range gates, I would expect the phase differences between the measurements at each range gate and the REWS at the rotor plane to appear in the equation. To better understand Eq. 32, as well as Eq. 29 for $S_{LL}(k_1)$, can you explain the assumptions in the derivations in more detail? For example, is the cross-spectrum in Eq. 32 derived assuming the lidar-estimated REWS is delayed in time according to Taylor's hypothesis so it is in phase with the REWS at the rotor plane? Similarly, when averaging lidar measurements at different range gates (Eq. 28), do you delay the measurements at different range gates in time so they are in phase with the nearest range gate, according to Taylor's hypothesis, before averaging? If not, how is the phase delay between measurements at different range gates accounted for in Eq. 32 (it seems like it is already included in Eq. 29)?

Thanks a lot for the reviewer's comments and suggestions. We have added text in Ln 313 to specify the assumptions we have made when deriving Equation 32. As for Equation 29, we kept the Equation but explain that the data is phase shifted in practical LAC so that there is no phase shift caused by the longitudinal seperations. Please see Ln 298.

7. Reviewer comment 1: Line 330: "This also indicates that the filter design is not sensitive to the change in turbulence parameters. . . and a constant filter design is robust." How does the filter design depend on the wind speed? Do the cutoff frequencies change? Author response: Thanks for pointing this out, we have removed Table 5 and added Figure 5, which shows the cutoff frequencies as a function of the mean wind speed. We also added the discussions around the line 340 as: "The cutoff frequencies as a function of mean wind speed are calculated by fitting the GRL and are shown in Figure 5. Firstly, both turbulence models indicate that the cutoff frequencies depend on the mean wind speed linearly. Therefore, the cutoff frequency of the filter can be scheduled based on this linearity"

Reviewer comment 2: When varying the wind speed to determine the cutoff frequencies, are you keeping the TI constant or changing it according to the IEC standard?

Thanks a lot for the reviewer's comments. We have clreaify this in Ln 354. We indeed adjusted the TI by the mean wind speed according to IEC standard.

8. Reviewer comment 1: Line 546: "the electricity productions are similar either using LAC or not. . . " Again, there is a significant drop in power at 14 m/s with LAC. What causes this? Author response: Thanks for your comment. The reason of the lower mean power at 14m/s has been explained in 42. We now analyzed the reason for power drop at 14m/s in line 582. However, we did not get the reviews

opinion that the EP is reduced with LAC. In the plot, the right side y axis is the relative reduction compared to FB-only control. If it is negative value, it means the FFFB gives higher value compared to FB-only control. So the EP is actually increased (very slightly) with LAC. However, since the increment is marginal, our conclusion is that LAC has marginal impact on the EP.

Reviewer comment 2: My mistake. I misinterpreted the meaning of the negative reduction in energy production in the plots. Thanks for your feedback.

9. Reviewer comment 1: In many places throughout the manuscript, there are citations without parentheses, for example line 44: "Mann (1994)." If the reference is actively used as part of the sentence, it is ok to leave the parentheses out, such as lines 46-48. Otherwise, I suggest using parentheses, for example, as is done in line 25. Author response: Thanks for the careful suggestion. We have went through the paper and corrected relative citations.

Reviewer comment 2: The citations have been improved significantly. There might be a few that still are missing parentheses, however. For example, Ln. 315: "Schlipf et al." Thanks for your feedback, we have checked again the citations through the manuscript.

10. Reviewer comment 1: Line 315: "If a filter with the gain. . . " This sentence is hard to understand and appears to be incomplete. Author response: Thanks for pointing this out, this sentence is now rewritten as: "If a filter is designed to have a gain of GRL(f ), it turns out to be an optimal Wiener filter (Simley and Pao (2013), Wiener et al. (1964)), which results in minimal output variance for a multi-inputs multi-outputs system."

Reviewer comment 2: The phrase "results in minimal output variance for a multi-inputs multi-outputs system" could use some explanation. What does this mean in the context of the LAC application, and what specific variance does the filter minimize in this application? Thanks for your feedback, we have added the explanation in Ln 345 as: "For example, in LAC, if the system is modeled as a system with two inputs: REWS and lidar-estimated REWS, and one output: rotor speed, the Wiener filter leads to minimal rotor speed variance (Simley and Pao, 2013).

**New Comments:**

1. Ln. 87: Consider providing a little more information about ROSCO here. For example, that it is a reference controller representing an industry standard control system.

Thanks for your feedback, we have added the explanation in Ln 61 as: "ROSCO is an open, modular, and fully adaptable baseline wind turbine controller with industry-standard functionality.

2. Ln. 206: "In this work, we emphasize analyzing the impact of turbulence length scale on turbine loads and LAC benefits": Would it be more accurate to say that you are analyzing the impact of turbulence length and the Gamma anisotropy parameter as well, since Gamma is different for the three stability classes?

Thanks for your feedback, we have added the anisotropy in the sentence.

3. Ln. 220: Please define the quad-coherence

Thanks for your suggestion, we have added the definition in Ln 144.

4. Ln. 225: "In their study, a smaller intercept was found for a more stable class. Also, Simley and Pao (2015) studied the turbulence...": It would be worth discussing whether the longitudinal coherence from the modified Mann model also shows different intercepts for different stability classes, even though the Mann model may not explicitly have an intercept parameter like $b_x$.

Thanks for the comment, actually we have shown these in Figure 2(c). The blue lines are the longitudinal coherence from the three stability conditions. It can be seen that the blue lines have a lower interception than the other two lines. To make a connection to the figure, we have added a bracket in the end of the sentence to indicate that the results are shwon in Figure 2.

5. Figure 2: It would help to state what mean wind speed these spectra are generated for.

Thanks for your suggestion, we have added the mean wind speed in the Figure caption.

6. Section 3.2: I think more details about how the lidar-estimated REWS is formed should be provided here to better understand the spectrum calculations. In particular, how do you combine measurements at different range gates? Do you delay the measurements from the farther range gates according to Taylor's hypothesis so they are in phase with the measurements from the closest range gate before averaging? Or do you average all measurements at the same time? This decision should affect the spectrum equation in Eq. 29.

Thanks for your suggestion, we have added more discussions after Ln 300.

7. Eq. 28: Is this equation missing a negative sign? According to the angle definitions in Fig. 3, cos(phi) will be negative. If the LOS velocity $v_{los,i}$ is positive, then $u_{LL}$ will be negative as written.

Thanks for your question. Indeed the first element of the unit vector is negative. In our implementation, we always use the inertial coordinate system shown in Figure 3 and use Equation 27 to calculate the LOS speed. So LOS speed is also negative. Then, the $u_{LL}$ will be positive.

8. Figure 4: Can you list what wind speed these coherence and transfer function curves are generated for? Thanks for your suggestion, we have added the mean wind speed in the Figure caption.

9. Ln. 403: "$M_g = P_{\mathrm{rated}}/(\eta\Omega_{\mathrm{gf}})$": the way this is written, it is unclear whether $\Omega_{\mathrm{gf}}$) is in the numerator or denominator. This comments applies to line 503 as well.

Thanks for your careful reading, we have added a bracket to correct the formulas.

10. Ln. 412: "lidar-assisted pitch forward signal": Be consistent about "forward" vs. "feedforward" throughout the text.

*Thanks for your careful reading, we have searched whole text to unify the wording.*

11. Ln. 419: "where $f_c$ is the cutoff frequency": I think it would help to mention that this is the same cutoff frequency that is discussed in Sect. 3.4.

*Thanks for your suggestion, we have added the connection.*

12. Ln. 420: "The pitch forward signal is then sent to ROSCO after accounting for the pitch actuator delay...": Should the filter delay $T_{filter}$ also be mentioned in this sentence?

*Thanks for your suggestion. You are right. We have added the filter delay.*

13. Ln. 421: "and the half of the time averaging window". Consider clarifying by saying this is "half of the lidar scan time averaging window" or similar.

*Thanks for your suggestion. We have added definition that time averaging window equals to the lidar full scan time.*

14. Ln. 425: "...is chosen to be half of the time averaging window". Why is $T_{window}$ set to half the time averaging window? Doesn't the factor of one half already appear in Eq. 40, meaning $T_{window}$ should be the full averaging time window?

*Thanks for your careful reading. It was a mistake and we have corrected this. The new sentence is:Here, Twindow=1 s is the time averaging window equivalent to one full scan time tof the lidar. It is multiplied by 1/2 in Equation 42, because of the phase delay property of the time averaging filter (Lee et al., 2018).*

15. Ln. 457: "Each 4D turbulence field has a size of 4096 x 11 x 64 x 64 grid points...": Can you mention the time resolution here?

*Thanks for your suggestion. We have added the time step (0.5s) in Ln 489.*

16. Ln. 616: "The electrical power generation is not affected by introducing LAC.": There is a small change, so perhaps "not significantly affected" would be more appropriate.

*Thanks for your suggestion. We have added 'not significantly'.*

17. Ln. 669: "also reduces the variation in rotor speed, pitch rate, and electrical power clearly": Why are reductions in pitch rate mentioned here for the Kaimal model, but not on line 664 for the Mann model?

*Thanks for your careful reading. That was a mistake and we have added it for Mann model.*

18. Ln. 675: "Overall, we found the benefits of lidar-assisted control by the Kaimal model are slightly different from the results obtained using the Mann model.":

Are there any interesting differences to mention here?
Thanks for your suggestion. We added more discussion as: The benefits of lidar-assisted control simulated using the Mann model is slightly better than that using the Kamal model, which can be caused by differences in the turbulence spatial coherence between two models. The lidar preview quality modeled using the Mann model is generally superior to that modeled using the Kaimal model.

New minor comments:
1. Ln. 57: "acting" − > "acts"? Thanks for your suggestion. We corrected this in the second revision.

2. Ln. 61: This may be a personal preference, but I think it is helpful to spell out acronyms like "ROSCO" in the body of the text the first time, even if they are defined in the abstract as well. Thanks for your suggestion. We corrected this in the second revision.

3. Ln. 63: can you provide references for FAST and OpenFAST? Thanks for your suggestion. We added references.

4. Ln. 67: "tool" − > "tools"? Thanks for your suggestion. We corrected this in the second revision.

5. Figure 1 caption: Be consistent on the use of "pulsed lidar" vs. "pulse lidar" throughout the paper. Thanks for your suggestion. We corrected this in the second revision.

6. Ln. 71: "value" − > "values"? Thanks for your suggestion. We corrected this in the second revision.

7. Ln. 140: "interested" − > "interesting"? Thanks for your suggestion. We corrected this in the second revision.

8. Ln. 244: "gives large values of $a_x$": To make the point more clear, should this say something like "gives unrealistically large values of $a_x$"? Thanks for your suggestion. We corrected this in the second revision.

9. Ln. 380: "propriety" − > "property"? Thanks for your suggestion. We corrected this in the second revision.

10. Ln. 501: "even the constant" − > "even though the constant"? Thanks for your suggestion. We corrected this in the second revision.

11. Ln. 503: "$M_g = P_{\text{rated}}/(\eta \omega_{\text{gf}})$": In section 4.3, the generator speed is written as capital Omega. Should it be the same here? Thanks for your suggestion. We

corrected this in the second revision.

12. Ln. 511: "less low-frequency rotor speed fluctuation": This is a little confusing. Consider rephrasing as "reduced low-frequency rotor speed fluctuations" or similar. Thanks for your suggestion. We corrected this in the second revision.

13. Ln. 521: "RWES" − >"REWS" Thanks for your suggestion. We corrected this in the second revision.

14. Ln. 531: "most interested" − > "most interesting"? Thanks for your suggestion. We corrected this in the second revision.

15. Ln. 544: "1 p": Usually I see this written with a capital "P" Thanks for your suggestion. We added definition of 1p and 3p but kept lower case.

16. Ln. 556: "statics" − > "statistics" Thanks for your suggestion. We corrected this in the second revision.

---

## Author Response (AR3)

**Authors' response to Review 2 (3nd revision):**
**Evaluation of lidar-assisted wind turbine control under various turbulence characteristics**

Feng Guo*, David Schlipf and Po Wen Cheng

02.01.2023

We would like to sincerely thank all the reviewers for taking their valuable time to read our manuscript and provide constructive comments. Special thanks to Review 2 for your careful review of the paper and further constructive comments in the third round of review.

Please find below our response to the reviewer 2's comments. The reviewer's comments in the report are repeated in black text, our response should be given in blue text, and if necessary, the corresponding corrections are provided in red text.

**Response to comments of Anonymous Referee #2 (3nd round)**

**Overall comments**

The manuscript has been greatly improved and almost all of my comments have been addressed. I just have a few last comments I hope the authors can address.

We are very appreciative of the referee for the positive feedback after the second round of revision on the manuscript.

3. Reviewer comment 1: Eq. 20: Why is the real number operator needed here? By definition, won't the coherence be a positive real number? Otherwise, can you explain how coh11 can contain imaginary components? Author response: Thanks for the your comments. Indeed, the magnitude squared coherence is real and positive. In terms of the least square fitting in Equation (21) (previously 20), we are fitting the co-coherence. We have corrected the equation now and added Equation (10) to explain the definition of co-coherence. Reviewer comment 2: Can you explain why you are fitting the co-coherence instead of the magnitude squared coherence? Author response 2: Thanks a lot for the reviewer's comments. We chose to fit the co-coherence because the exponential coherence model (used for Kaimal spectra) is a real function and it only has co-coherence. We have added explanations in Ln 220. Reviewer comment 3: This makes sense. But I suggest also clarifying that this is necessary because the yz-plane coherence from the Mann model includes quad-coherence, for example "...because the exponential coherence model (Equations 13 and 19) only includes the real co-coherence, whereas the Mann coherence model includes both co-coherence and quad-coherence.", unless if this has been discussed earlier.

Thanks a lot for the suggestion, we have added the sentence as : whereas the Mann coherence model includes both co-coherence and quad-coherence.

10. Reviewer comment 1: Line 315: "If a filter with the gain. . . " This sentence is hard to understand and appears to be incomplete. Author response: Thanks for pointing this out, this sentence is now rewritten as: "If a filter is designed to have a gain of GRL(f ), it turns out to be an optimal Wiener filter (Simley and Pao (2013), Wiener et al. (1964)), which results in minimal output variance for a multi-inputs multi-outputs system." Reviewer comment 2: The phrase "results in minimal output variance for a multi-inputs multi-outputs system" could use some explanation. What does this mean in the context of the LAC application, and what specific variance does the filter minimize in this application? Author response 2: Thanks for your feedback, we have added the explanation in Ln 345 as: "For example, in LAC, if the system is modeled as a system with two inputs: REWS and lidar-estimated REWS, and one output: rotor speed, the Wiener filter leads to minimal rotor speed variance (Simley and Pao, 2013).

Reviewer comment 3: The Wiener filter minimizes the mean square error between a signal and the estimate of the signal, so I still don't understand the phrase "results in minimal output variance for a multi-inputs multi-outputs system." Is there a reference that explains this idea more? Or is this true if the Wiener filter is used in a specific way in the MIMO system? Further, for the example provided from Simley and Pao, 2013 with a system with two inputs: REWS and lidar-estimated REWS, and one output: rotor speed, the reason the Wiener filter minimizes rotor speed variance is that the problem is formulated so that the output rotor speed is a function of the difference between the REWS and the lidar-estimated REWS. I don't know if using the Wiener filter would, in general, lead to minimizing output variance for MIMO systems.

We appreciate that the reviewer points out his or her detailed thoughts, we totally agree with the reviewer. Indeed, it was because Simley and Pao formulated the problem by using the Wiener filter for a specific MIMO system so that it results in results in minimal output variance for the MIMO system. A single Wiener filter, by definition, only produces an estimate of a desired or target signal and minimizes the mean square error between a signal and the estimate of the signal. We have now rewritten the sentences as: If a filter is designed to have a gain of $G_{\mathrm{RL}}(f)$, it turns out to be an optimal Wiener filter (Simley and Pao, 2013; Wiener et al.,1964), which results produces an estimate of a desired or target signal (here the $u_{\mathrm{RL}}$). The Wiener filter minimizes the mean square error between the target signal and the estimate of the signal. When used for LAC, if the system is modeled as a system with two inputs: REWS and lidar-estimated REWS, and one output: rotor speed, the Wiener filter leads to minimal rotor speed variance as formulated by Simley and Pao, (2013).

6 (from New Comments section). Reviewer comment 1: Section 3.2: I think more details about how the lidar-estimated REWS is formed should be provided here to better understand the spectrum calculations. In particular, how do you combine measurements at different range gates? Do you delay the measurements from the farther range gates according to Taylor's hypothesis so they are in phase with the measurements from the closest range gate before averaging? Or do you average all measurements at the same time? This decision should affect the spectrum equation in Eq. 29. Author comment 1: Thanks for your suggestion, we have added more discussions after Ln 300.

Reviewer comment 2: One small comment is that in Eq. 32, I think the sign of the time shift might be wrong. According to the sign convention for the "$x$" dimension shown in Fig. 3, ($x_i$ - $x_{\mathrm{nrg}}$) will be negative, leading to a shift forward in time. But I think the correct time shift for range gates farther away from the turbine should be a shift backward in time when adding the measurements from different range gates. Should the numerator of the time shift term actually be ($x_{\mathrm{nrg}}$ - $x_i$)?

We appreciate the reviewer's careful reading and review. The reviewer is correct and it should be $x_{\mathrm{nrg}}$ - $x_i$. We have made corrections in the revised version.